



# Spatial variations in sedimentary N-transformation rates in the North Sea (German Bight)

Alexander Bratek[1,2], Justus van Beusekom[1,3], Andreas Neumann[1], Tina Sanders[1], Jana Friedrich[1], Kay-Christian Emeis[1,2] and Kirstin Dähnke*[1]

[1] Helmholtz-Zentrum Geesthacht, Institute for Coastal Research, Geesthacht, Germany
[2] University of Hamburg, Center for Earth System Research and Sustainability, Institute for Geology, Hamburg, Germany
[3] University of Hamburg, Center for Earth System Research and Sustainability, Institute for Hydrobiology and Fisheries, Hamburg, Germany

*Correspondence to:* kirstin.daehnke@hzg.de

**Abstract**

In this study, we investigate the role of sedimentary N cycling in the Southern North Sea. We present a budget of ammonification, nitrification and sedimentary $NO_3^-$ consumption / denitrification in contrasting sediment types of the German Bight (Southern North Sea), including novel net ammonification rates. Dissolved inorganic nitrogen concentration (nitrate, nitrite and ammonium) in the water column showed low levels between 0.2 to 3.2 µmol $L^{-1}$. We incubated sediment cores with labeled nitrate and ammonium to calculate net and gross N transformation rates. The results show that impermeable sediments are the main site of ammonification (on average 10.2 ±1.2 mmol $m^{-2}$ $d^{-1}$) and that they are an important source for primary producers in the water column, contributing ~17 to 61 % of reactive nitrogen in the water column. Ammonification and oxygen penetration depth are the main drivers of sedimentary nitrification. One third of freshly produced nitrate in impermeable sediment and two-thirds in permeable sediment were reduced to $N_2$. The semi-permeable and permeable sediments are responsible for ~80 % of the total benthic $N_2$ production rates (~890 t N $d^{-1}$) in the southern North Sea. We conclude that impermeable sediments are important sources of reactive N and that semi-permeable and permeable sediments are the main sinks of reactive N, counteracting eutrophication in the southern North Sea (German Bight).





## 1 Introduction

The continental shelves and coastal margins make up for <9 % of the total area of ocean surface, but are responsible for vast majority of the biogeochemical cycling both in the water column and in the sediments (Jorgensen, 1983). For instance, 30 % of global marine primary production occurs in coastal, estuarine and shelf systems (LOICZ, 1995), and nutrient regulation in shelf sediments is a particularly valuable ecosystem service (Costanza et al., 1997).

The German Bight is part of the southern North Sea and is bordered by densely populated and industrialized countries, and receives large amounts of nutrients via river discharge (e.g., Rhine, Maas, Elbe, Weser, Ems) (Los et al., 2014). This caused clear eutrophication symptoms such as phytoplankton blooms, oxygen deficiencies and macrobenthos kills especially during the 1980s (Hickel et al., 1993; von Westernhagen et al., 1986) in the North Sea. In the adjacent Wadden Sea intense phytoplankton blooms, a decrease of seagrass and massive blooms of opportunistic macroalgae were attributed to eutrophication (Cadée and Hegemann, 2002; Dolch et al., 2013; Reise and Kohlus, 2007; Reise and Siebert, 1994). Since the mid 1980s, the nitrogen (N) loads into the German Bight have been decreasing, but the entire SE North Sea is still flagged as an eutrophication problem area (OSPAR, 2010).

Nitrogen availability increases primary production on a variety of spatial and temporal scales. At present, major nitrogen sources for the Southern North Sea are agricultural and urban waste water, and to a lesser extent, a variety of reactive N emission (e.g., nitrogen oxides from burning fossil) (Emeis et al., 2015).

Internal N cycling in sediments (e.g., assimilation, ammonification and nitrification) change the distribution and speciation of fixed N, but not the overall amount of N available for primary production (Casciotti, 2016). Removal of $NO_3^-$ through denitrification and anammox in anoxic conditions back to unreactive $N_2$, however, does remove N from the biogeochemical cycle (Neumann et al., 2017).

Because these eliminating processes are confined to suboxic and anoxic conditions, they occur in sediments in the generally oxygenated North Sea. In spite of their putative relevance as an ecosystem service, very little is known about N cycling and N transformation rates in the sediment. This is in part due to the complexity created by coupled ammonification-nitrification in which different N processes, such as assimilation and denitrification, interact and affect the $NH_4^+$ and $NO_3^-$ concentrations in pore waters. To our knowledge, no ammonification rates in the North Sea have been quantified, whereas nitrification rates in permeable sediments were found to be in the same order of magnitude as denitrification rates (<0.1 to ~3.0 mmol $m^{-2}$ $d^{-1}$, Tab. 1) (Marchant et al., 2016). N loss in the German Bight has been studied by several authors (Deek et al., 2013; Lohse et al., 1993; Marchant et al., 2016; Neubacher et al., 2012; Neumann et al., 2017) showing high spatial, temporal and seasonal variability.



The main N loss process in the North Sea is denitrification, whereas and anammox plays a minor role (Bale et al.,
2014; Marchant et al., 2016). The main drivers of denitrification are organic matter content and permeability of
the sediment (Neumann, 2012), and recent studies suggest that permeable sediments account for about 90
% of the total benthic $NO_3^-$ consumption in the German Bight (Neumann et al., 2017).
Quantifying N dynamics based solely on changes in N concentrations provides limited insight into underlying
reactions, as only net changes can be observed. Previous authors used different methods for determination of
specific N rates. Lohse et al. (1993) used the acetylene block method, core flux incubations and isotope pairing in
the early 1990s types to determine denitrification rates in a variety of sediment types (Tab. 1). Deek and co-authors
(2013; 2011) investigated N-turnover in the Wadden Sea and in the extended Elbe estuary using core flux
incubations and isotope pairing. Marchant et al. (2016) measured denitrification rates in permeable sediments
obtained from slurry incubations and percolated sediment cores. More recently, Neumann et al. (2017) used pore-
water $NO_3^-$ concentration gradient profiles to determine $NO_3^-$ consumption rates in the German Bight.
Stable isotope techniques offer several approaches to quantify N turnover processes, and $^{15}N$ tracer studies have
been widely used to determine N transformation rates (e.g. nitrification and denitrification) (Brase et al., 2018;
Deutsch et al., 2009; Henriksen and Kemp, 1988; Sanders et al., 2018; Wankel et al., 2011). In this study, we use
an isotope dilution method that can unravel several N-processes like ammonification, assimilation, nitrification,
denitrification, dissimilatory $NO_3^-$ reduction to $NH_4^+$ (DNRA) and sedimentary $NO_3^-$ consumption / $N_2$ production
within sediments. $^{15}N$ dilution of $NH_4^+$ and $NO_3^-$ (Koike and Hattori, 1978; Nishio et al., 2001b) can be used to
estimate gross N transformation rates by measuring the isotopic dilution of the substrate and product pools,
respectively (Burger and Jackson, 2003; Hart et al., 1994; Ward, 2008). The $^{15}N$ dilution method accounts for
changes in both N pool size and $^{15}N$ enrichment during a short sampling interval. This method has four main
advantages over balancing sediment-water exchanges in other studies: (1) The appearance of $^{15}N$ in the $NH_4^+$ pool
during the incubation allows an estimate of ammonification rates, (2) the isotopic dilution of $NO_3^-$ tracks
nitrification rates, (3) labeling of $N_2$ for an estimate of $N_2$ production (Holtappels et al., 2011) and (4) the detection
of assimilation rates.
This study is conducted within the project "North Sea Observation and Assessment of Habitats" (NOAH). One
important aspect of the project is to investigate the biogeochemical status and functions of the sea floor, especially
nitrogen cycling, to gauge the eutrophication mitigation potential in light of continuing high human pressures
(https://www.noah-project.de).
In this paper, we investigate internal N rates of ammonification, nitrification and $NO_3^-$ consumption /
denitrification at four stations across sediment types (clay/silt, fine sand, coarse sand) in the German Bight (North



Sea) during late summer (August/September) 2016. To assess the internal sediment N processes and the rates of
reactive N release to the water column, we incubated sediment cores amended with $^{15}NH_4^+$ and $^{15}NO_3^-$. We quantify
the benthic gross and net N transformation rates and evaluate the environmental controls underlying spatial
variabilities. We further evaluate the role of ammonification as a source of reactive nitrogen for primary producers,
of nitrification and of denitrification in the Southern North Sea.
**2 Material and Methods**
**2.1 Study site**
Sediment samples were taken in the German Bight (Southern North Sea), an area that is strongly influenced by
nutrient inputs from large continental rivers. The salinity in the coastal zone of the North Sea ranges between ~30
and 35, and the average flushing time is 33 days (Lenhart and Pohlmann, 1997).
The sampling sites are part of the NOAH (North Sea Assessment of Habitats) assessment scheme (Fig. 1). The
sites represent typical sediment types based on statistics of granulometric properties, organic matter content,
permeability, and water depth (https://doi.org/10.1594/PANGAEA.846041).
**2.2 Sampling and core incubation**
The sampling was performed in August and September 2016 during R/V *Heincke* cruise HE-471 in the German
Bight (Fig. 1). The water depth at the sites varied between 25.2 m (NOAH-C) to 36.0 m (NOAH-D) (Tab. 2). At
each station, the water column was sampled at five depths with a rosette sampler equipped with Niskin bottles, a
CTD, and chlorophyll and $O_2$ sensors. For the nutrient analysis, water samples were filtered thought a 25-mm
diameter glass fiber filter (GF/F, Sartorius, 0.7 µm nominal pore size) and frozen immediately at -20 °C.
From each station, sediment multicores equipped with acrylic tubes (PMA) with an inner diameter of 10 cm and a
length of 60 cm were recovered and four intact sediment cores from each station (exception: Station NOAH-D,
only 3 cores could successfully be retrieved) were incubated in a gas tight batch-incubation setup for 24 hours.
Overlying water was stirred with a magnetic stirrer coupled to an external rotating magnet. The water temperature
was held constant at *in situ* conditions (~19 °C). Water temperature and oxygen concentration of the overlying
water of each sediment core were measured continuously with optodes (PyroScience, Germany.
Two sediment cores (Station NOAH-D 1 core only) were enriched with $^{15}NH_4^+$ (50 at-%, manufacturer), the other
two cores were amended with $^{15}NO_3^-$ (50 at-%, manufacturer). $NH_4^+$ and $NO_3^-$ concentration of the added tracer
solution was the same as the bottom water concentrations (Tab. 2).The label addition was calculated aiming for a
maximum enrichment of 5.000 ‰ in substrates and products, because higher delta-values influence the accuracy
of the mass spectrometer.



Samples were taken every 6 hours. Upon sampling, incubation water was filtered with a syringe filter (material,
manufacturer, 0.45 µm pore size) and frozen in exetainers (11.8 ml, Labco, High Wycombe, UK) at -20 °C for
later analyses of nutrients and stable isotope signatures ($\delta^{15}NH_4^+$, $\delta^{15}NO_3^-$). Additional samples for the analyses of
dissolved nitrogen ($N_2$) were taken without filtration, and were preserved in exetainers (5.9 ml, Labco, High
Wycombe, UK) containing 2 % of a $ZnCl_2$ solution (1 M). Samples were stored at 4 °C under water until analysis.
**2.3 Analyses**
**Dissolved inorganic nitrogen concentrations**
$NO_x$, $NO_2^-$ and $NH_4^+$ concentrations of the water column samples were determined in replicate with a continuous
flow analyzer (AA3, Seal Analytics, Germany) according to standard colorimetric techniques ($NO_x$, $NO_2$:
(Grasshoff et al., 1999), $NH_4^+$: (Kérouel and Aminot, 1997)). $NO_3^-$ concentration was calculated by difference of
$NO_x$ and $NO_2^-$. Based on replicate analyses, measurement precision for $NO_x$ and $NO_2^-$ was better than 0.1 µmol L$^-$
$^1$ and better than 0.2 µmol L$^{-1}$ for $NH_4^+$.
Water samples from core incubations were analyzed in duplicate for concentration of $NH_4^+$, $NO_2^-$ and $NO_3^-$ using
a multimode microplate reader Infinite F200 Pro and standard colorimetric techniques (Grasshoff et al., 1999) at
the ZMT, Bremen. The standard deviations were <1 µmol L$^{-1}$ for $NO_3^-$, <0.2 µmol L$^{-1}$ for $NO_2^-$ and <0.5 µmol L$^-$
$^1$ for $NH_4^+$.
**Nitrogen isotope analyses**
The nitrogen isotope ratios of $NO_3^-$ were determined via the denitrifier method (Casciotti et al., 2002; Sigman et
al., 2001). This method is based on the mass spectrometric measurement of isotopic ratios of $N_2O$ produced by the
bacterium *Pseudomonas aureofaciens*. Briefly, 20 nmoles of sample $NO_3^-$ were injected in a 20 ml vial containing
MilliQ. Two international standards were used (IAEA-$NO_3^-$ $\delta^{15}N$ = +4.7 ‰, USGS-34 $\delta^{15}N$ = -1.8 ‰) for a
regression-based correction of isotope values. For further quality assurance, an internal standard was measured
with each batch of samples. The standard deviation for $\delta^{15}N$ was better than <0.2 ‰
For ammonium isotope measurements, nitrite was removed by reduction with sulfamic acid (Granger and Sigman,
2009) before $NH_4^+$ was chemically oxidized to $NO_2^-$ by hypobromite at pH ~12 and then reduced to $N_2O$ using
sodium azide (Zhang et al., 2007). 10 nmol of $NH_4^+$ were injected, and all samples with [$NH_4^+$] >1 µmol L$^{-1}$ were
analyzed. For the calibration of the ammonium isotopes, we used three international standards (IAEA-N1 $\delta^{15}N$ =
+0.4 ‰, USGS 25 $\delta^{15}N$ = -30.4 ‰, USGS 26 $\delta^{15}N$ = +53.7 ‰). The standard deviations were better than 1 ‰.
$N_2O$ produced either by the denitrifier method or the chemical conversion of ammonium was analysed with a
GasBench II, coupled to an isotope ratio mass spectrometer (Delta Plus XP, Thermo Fisher Scientific).
**Membrane inlet mass spectrometry**



$N_2$ production was measured by a membrane inlet mass spectrometer (MIMS, inProcess Instruments), which
quantifies changes in dissolved $N_2$:Ar ratios (Kana et al., 1994). During the measurements, the water samples were
maintained in a temperature-controlled water bath (16 °C). For calibration, we measured 4 salinities, from 0 to 35
after each 10th water sample. We measured the production of $^{28}N$, $^{29}N$ and $^{30}N$ to quantify the $N_2$ production. Due
to the low labeling percentage, a distinction of anammox and denitrification rates was not possible. The internal
precision of the samples was <0.05 % for $N_2/A_r$ analyses.
**Sediment samples**
The surface sediment samples of the cruises HE 383 (06/07.2012) and HE 447 (06.2015) for NOAH-D were
analyzed for total carbon and total nitrogen contents with an elemental analyzer (Carlo Erba NA 1500) via gas
chromatography calibrated against acetanilide. The total organic carbon content was analyzed after removal of
inorganic carbon using 1 mol $L^{-1}$ hydrochloric acid. The standard deviation of sediment samples was better than
0.6 % for $C_{org}$ and 0.08 % for N determination.
**Respiration and transformation rates**
_Net process rates_
The oxygen consumption, net rates of ammonification, nitrification and denitrification were calculated based on
concentration changes in the sediment incubations. The respective net process rates were calculated as follows:
$$r_{net} = d(C)*V/d(t)*A \;[\text{mmol m}^{-2}\,\text{d}^{-1}] \tag{1}$$
where $d(C)$ is the oxygen, nutrient or the nitrogen ($N_2$) concentration at the start and at the end of the experiment,
$V$ is the volume of the overlying water, $d(t)$ is the incubation time and $A$ is the surface area of the sediment.
_Gross rates of ammonification, nitrification and assimilation_
Gross rates of ammonification and nitrification ($r_{gross}$) were calculated based on $^{15}N$ isotope dilution (Koike and
Hattori, 1978; Nishio et al., 2001a), i.e, ammonification rates are calculated based on $^{15}NH_4^+$ additions, nitrification
rates are based on $^{15}NO_3^-$ additions:
$$r_{gross} = [\ln(f^{15}N_{end}/f^{15}N_{start})]/[\ln(C_{end}/C_{start})]*(C_{start}-C_{end}/t)*(V/A*\Delta t) \tag{2}$$
where $C_{start}$ is the initial $NH_4^+$ or $NO_3^-$ concentration, $C_{end}$ is the concentration at time t, and $f^{15}N_{start}$ and $f^{15}N_{end}$
represent $^{15}N$ atom% excess (Brase et al., 2018), $V$ is the volume of the overlying water and $A$ is the surface area
of the sediment. All rates are given in mmol $m^{-2}$ $d^{-1}$
Based on these calculations, we derived $NH_4^+$ assimilation as follows:
$$rNH_4^+{}_{ass} = rNH_4^+{}_{gross} - rNH_4^+{}_{net} - rNitr_{gross} \;[\text{mmol m}_{-2}\,\text{d}^{-1}] \tag{3}$$
where $rNH_4^+{}_{gross}$ is the gross ammonification rate, $rNH_4^+{}_{net}$ is the net ammonification rate and $rNitr_{gross}$ represents
the gross nitrification rate.



**Oxygen penetration depth**
The oxygen penetration depth in the sediment of each station were measured using microoptodes (50 µm tip size;
Presens, Germany). The optodes were moved vertically into the sediment with a micromanipulator (PyroScience,
Germany), in steps of 100-200 µm, depending on the oxygen concentration. Three $O_2$ profiles were measured in
one sediment core of each station.
**3 Results**
**3.1 DIN concentrations in the water column**
$NO_3^-$ concentrations in the water column were low at all stations (0.1 µmol $L^{-1}$ or lower, Tab. 3). $NO_2^-$
concentrations were low at the permeable sediment stations NOAH-A, NOAH-D and NOAH-E with ($\leq$0.1 µmol
$L^{-1}$ below the thermocline). At the impermeable sediment station (NOAH-C), $NO_2^-$ concentration was 0.7 µmol $L^-$
$^1$. NOAH-C had also highest $NH_4^+$ concentrations with 2.0 $\pm$0.2 µmol $L^{-1}$, whereas $NH_4^+$ concentrations at the
permeable sediment stations were lower (0.3 to 0.8 µmol $L^{-1}$).
**3.2 Benthic oxygen fluxes**
The $O_2$-fluxes from the water column into the sediment (here: negative fluxes) vary between individual cores and
sampling station. The lowest oxygen flux was determined at the permeable sediment station NOAH-A with -10.0
mmol $m^{-2}$ $d^{-1}$ (Fig. 2), the highest oxygen flux was measured at the impermeable sediment station NOAH-C with
-53 mmol $m^{-2}$ $d^{-1}$. The semi-permeable sediment station NOAH-D had an oxygen flux of -18.5 to -30.6 mmol $m^{-2}$
$d^{-1}$.
**3.3 Nitrogen transformation rates**
*Ammonification*
The highest net and gross ammonification rates were measured in the impermeable, organic-rich sediment at
station NOAH-C (6.8 $\pm$2.3 mmol $m^{-2}$ $d^{-1}$ and 8.3 $\pm$2.3 mmol $m^2$ $d^{-1}$ for net and gross ammonification, respectively;
Fig. 3 and Fig. 5).
The lowest ammonification rates were measured in the semi-impermeable sediment at station NOAH-D ($rNH_4^+{}_{net}$
$=0.5$ mmol $m^{-2}$ $d^{-1}$; $rNH_4^+{}_{gross} = 2.3$ $\pm$0.4 mmol $m^{-2}$ $d^{-1}$). The permeable sediment stations NOAH-A and NOAH-E
show ammonification rates of 2.4 $\pm$0.9 mmol $m^{-2}$ $d^{-1}$ and 3.6 $\pm$1.3 mmol $m^{-2}$ $d^{-1}$ (net and gross, respectively). Net
and gross ammonification rates are closely correlated ($r^2$=0.96; data not shown).
*Assimilation*
The $NH_4^+$ assimilation differed between stations, and ranged from <0.1 to 0.6 mmol $m^{-2}$ $d^{-1}$ (Fig. 3, Fig. 5). Rates
were lowest at the impermeable sediment station NOAH-C, and highest in the moderately permeable sediment at
station NOAH-D.





*Nitrification*
Net and gross nitrification rates varied significantly between stations. Net nitrification was highest at station
NOAH-C (impermeable sediment) and at station NOAH-D (semi-permeable sediment) with 0.9 ±0.7 and 1.0 ±0.3
mmol $m^{-2}$ $d^{-1}$, respectively (Fig. 3, Fig. 5). Gross nitrification was highest at NOAH-D (1.5 ±0.2 mmol $m^{-2}$ $d^{-1}$).
The lowest net (0.3 ±0.2 mmol $m^{-2}$ $d^{-1}$) and gross (0.7 ±0.4 mmol $m^{-2}$ $d^{-1}$) rates were observed in the permeable
sediment at station NOAH-A. Net and gross nitrification rates are closely correlated ($r^2$=0.75; Fig. 4).
*Nitrate consumption*
$NO_3^-$ consumption rates did not differ significantly between the stations and ranged from <0.1 to 0.8 mmol $m^{-2}$ $d^{-}$
$^1$ (Fig. 3, Fig. 5).
*Denitrification*
The denitrification rates of ranged from 0.4 to 2.4 mmol $m^{-2}$ $d^{-1}$ (Fig. 3, Fig. 5).The $N_2$ production rates in the
sediment of the stations did not vary significantly between stations. We found no indication of dissimilatory
nitrogen reduction to $NH_4^+$ (DNRA) in the sediment.
**Sedimentary organic matter descriptions**
The data show a clear correlation between sediment type and organic carbon and nitrogen content. Clay and silty
sediment (NOAH-C) have the highest organic carbon (0.73 %) and nitrogen (0.10 %) concentration (Tab. 2).
Medium sand stations (NOAH-A and NOAH-E) show the lowest $C_{org}$ (0.03 to 0.04 %) and total nitrogen (<0.01
to 0.01 %) concentrations.
**4 Discussion**
**4.1 Magnitude and relevance of ammonification**
A principal goal of this study was to assess for the first time the role of ammonification in the nitrogen cycle of
the German Bight. Ammonification releases $NH_4^+$ during the decomposition of organic matter and resupplies the
water-column inventory of reactive nitrogen. The quantification of gross ammonification rates is challenging,
because ammonium is readily assimilated by primary producers or is rapidly nitrified, so that ammonium
concentrations are often very low.
To the best of our knowledge, this study represents the first assessment of ammonification rates across typical
sediment types of the North Sea, covering a large range from 2.3 to 8.3 mmol $m^{-2}$ $d^{-1}$: Rates were mainly governed
by sediment texture and organic matter content. The impermeable muddy sediment at station NOAH-C with high
$C_{org}$ and TN content (0.73 % and 0.10 %, respectively, Tab. 2) had highest gross and net ammonification rates.
This is line with other studies showing enhanced rates in muddy coastal sediments (Caffrey, 1995; Mackin and
Swider, 1989; Nichols and Thompson, 1985; Sumi and Koike, 1990).



The sandy sediments at sites NOAH-A, NOAH-D and NOAH-E exhibited significantly lower gross
ammonification rates. This reflects the lower sediment organic matter content in these sandy sediments expressed
in $C_{org}$ (0.03 – 0.04 %) and N (0.01 – <0.01 %) concentrations (Caffrey, 1995), Tab. 2).
It is striking, though, that gross ammonification in the sandy sediment at station NOAH-E was almost twice that
of the other sandy stations NOAH-A and NOAH-D. There are two possible explanations for this enhanced
ammonium production, either (1) effects bioirrigation and bioturbation or (2) enhanced supply of organic matter
to the sediment surface. Station NOAH-E is located inside a pockmark field that had developed relatively recently,
between July and November 2015 (Krämer et al., 2017). Our assessment of OC and N content is based on samples
that were taken prior to the pockmark formation (Krämer et al., 2017)
(https://doi.org/10.1594/PANGAEA.883199). The sediment samples were taken from the depression inside an
individual pockmark, which was about ~0.2 deeper than the surrounding sediment (Krämer et al., 2017). It is
possible that organic matter from the water column accumulated in these transient structures, and that the organic
carbon and nitrogen content thus was elevated. A transient change in surface sediment composition, which is not
captured by our compositional data, may thus have caused the enhanced ammonification rate.
An alternative explanation is an elevation of ammonium fluxes from the sediment due to sediment reworking. In
the sediment incubations, we found a high benthic activity of *Spiophanes bombyx* and *Phoronis sp.*. Both benthic
organisms can increase the nutrient fluxes from the sediment to the bottom water, the oxygen penetration depth,
and, in turn, organic matter degradation in the oxic zone (Aller and Aller, 1998; Caffrey, 1995; Meysman et al.,
2006; van Amstel et al., 2007).
Under completely oxic conditions, the ratio of $NH_4^+$ release and $O_2$ consumption should approximate Redfield
ratios of about 1:8.6 (Thibodeau et al., 2010). Similar ratios were observed at the semi-permeable station NOAH-
D and in 2 of 4 sediment cores of the permeable station NOAH-A (Fig. 2), suggesting that in these cores most of
the organic matter was degraded under oxic conditions. At some sites (NOAH-C, NOAH-E), however, the $N:O_2$
ratio was much higher (1:7 to 1:2) than the Redfield ratio. Higher $N:O_2$ ratios may be partly related to the quality
of the organic matter: (Hargrave et al., 1993) measured also higher ammonium fluxes relative to oxygen
consumption in North American east coast sediments. They speculated that remineralisation of organic nitrogen
is faster than that of organic carbon.
We presume that the enhanced production of ammonium relative to $O_2$ consumption reflects the importance of
anoxic ammonium generation, i.e., during methanogenesis or sulfate reduction (Jorgensen, 1982; Jorgensen et al.,
1990; Kristensen et al., 2000; Miyajima et al., 1997). This is especially evident at station NOAH-C, where oxygen



penetration depth in the impermeable, organic-rich sediment is lowest, and where increasing $NH_4^+$ concentrations
with depth indicate decomposition or organic matter in the absence of free oxygen (Hartmann et al., 1973).
Sedimentary ammonium production and fluxes of ammonium into the water column contribute to water column
DIN concentrations. To assess the contribution of benthic ammonification to the water column N inventory, we
compared gross ammonification with the inventory of DIN below the thermocline. Assuming steady state, we find
a rapid turnover of sediment-derived DIN, in the order of ~<1-3 days (Tab. 3). This implies that even below the
thermocline, DIN is rapidly assimilated by phytoplankton. Previous publications showed that primary production
below the thermocline can amount to ~ 37 % of total primary production in the North Sea (van Leeuwen et al.,
2013; Weston et al., 2005). Assuming Redfield stoichiometry, our measured benthic $NH_4^+$ fluxes can support a
primary production of about 2.3 to 8.3 mmol $m^{-2}$ $d^{-1}$ or 0.2 – 0.6 g C $m^{-2}$ $day^{-1}$. This is in the lower range of
previously observed and modeled primary production rates in the North Sea during summer (Rick et al., 2006; van
Leeuwen et al., 2013; Weston et al., 2005). In total, though, we estimate that benthic N fluxes support between 13
% (at a water depth of 38 m) and 61 % at 10 m depth (Tab. 3) of primary production. This dependence of relative
sediment contribution on water depth has been observed previously for respiration processes (Heip et al., 1995).
Our data also match the calculation of Blackburn and Henriksen (1983) for Danish sediments, where N fluxes
could support 30-83 % of the nitrogen requirement of the planktonic primary producers (Blackburn and Henriksen,

288    1983).

In summary, our results show a rapid ammonification of organic matter and an intense benthic-pelagic coupling
during summer in the German Bight.
**4.2 Ammonia and nitrite oxidation (nitrification)**
Based on the interpolation of gross rates of ammonification, it is evident that ammonification contributes
significantly to nutrient regeneration in the German Bight. However, there is a clear difference between gross and
net ammonification rates, and beside ammonium assimilation, nitrification is an important ammonium sink.
Nitrification produces $NO_3^-$, which represents the largest DIN pool in the water column of the North Sea and is the
substrate for denitrification, and thus the link to an ultimate removal of fixed nitrogen from the water column.
We observed gross nitrification rates at all four stations ranging from 0.7 ±0.3 mmol $m^{-2}$ $d^{-1}$ at the sandy station
NOAH-A over 1.4 ±0.7 mmol $m^{-2}$ $d^{-1}$ in the impermeable sediment at station NOAH-C to 1.5 mmol $m^{-2}$ $d^{-1}$ in the
moderately permeable sediment at NOAH-D (Fig. 3, Fig. 4). Gross nitrification at the impermeable sediment
station NOAH-C accounted for around 16.2 % (±9.9 %), around 64.5 % (±9.1 %) at the semi-permeable station
(NOAH-D) and around 25.6 % (±11.4 %) at the permeable sediment stations of total DIN flux to the bottom water.
Overall, nitrification is in the same range as reported by Marchant et al. (2016) in sandy sediment near Helgoland



(0.2 to 3.0 mmol m$^{-2}$ d$^{-1}$; Tab. 1). We observed the highest net and gross release of NO$_3$$^-$ by nitrification at the
semi-permeable station NOAH-D, indicating that beside sediment texture, other processes affect the nitrification
rates (Marchant, 2014).
Nitrification rates are relatively independent of permeability, in contrast to ammonification. Instead, they were
negatively correlated (r$^2$ = 0.83) with oxygen penetration depth. The reactivity of organic matter and the bottom
water oxygenation affect the OPD and the nitrate gradient across the sediment-water-interface. High organic matter
reactivity will also lead to high diffusive nitrate fluxes (Alkhatib et al., 2012).
Nitrification rates are lowest at Station NOAH-A. Here, oxygen penetration depth is highest, and the sediment has
low organic matter content (Tab. 2), which obviously limits nitrification rates.
While individual correlations between C$_{org}$ or TN and nitrification are relatively weak, this indicates that organic
matter turnover indirectly controls nitrification rates. Generally, organic matter deposition in the sediment supports
higher ammonification rates, which in turn enhance nitrification under oxic conditions (Henriksen and Kemp,
1988; Rysgaard et al., 1996). Consequently, nitrification is affected by the NH$_4$$^+$ pool in the sediment, temperature,
salinity and O$_2$ (Henriksen and Kemp, 1988; Vouvé et al., 2000; Wankel et al., 2011).
Given these constraints, it is surprising that gross ammonification and gross nitrification rates are not correlated
(r$^2$ = 0.13). We suggest that this expresses a rate limitation of nitrifying bacteria. In sediments with high
ammonification rates and ammonium concentrations, ammonium oxidation is the limiting step for further
production of nitrate. Nitrifiers are slow-growing, with ammonium oxidation rates far below ammonification rates
(Kadlec and Wallace, 2009; Myers, 1975; Vymazal, 2010; Vymazal, 2007). Marchant et al. (2016) suggest that
other factors can additionally affect the rate of ammonia oxidation, such as the surface area available for microbial
colonization (Belser, 1979) or oxygen availability (Henriksen et al., 1993).
Overall, the gross NO$_3$$^-$ production (0.7 to 1.5 mmol m$^{-2}$ d$^{-1}$) was small relative to ammonification rates (2.3 to 8.3
mmol m$^{-2}$ d$^{-1}$). We find that nitrification is governed by a complex interplay of variables (ammonification rate,
sediment texture, permeability, organic matter availability and O$_2$ concentration) determine sediment reactivity as
reflected by oxygen penetration depth.
**4.3 Denitrification**
Denitrification, the reduction of NO$_3$$^-$ to gaseous N$_2$, reduces the pool of bioavailable N, and is therefore of great
interest in eutrophic coastal areas such as the southern North Sea. In our study, the measured denitrification rates
ranged from 0.4 to 2.4 mmol N m$^{-2}$ d$^{-1}$ (Fig. 3). This estimate of N$_2$ production is in line with other data from sites
in the German Bight estimated by either the isotope pairing technique or, as in our study, using isotope dilution



(Deek et al., 2013; Marchant et al., 2016) (Tab. 1). Our study covers more diverse sediment types, and thus allows
for an improved extrapolation of rates to the total German Bight area.
Variations in denitrification can be attributed to seasonal variations in oxygen supply, changing bottom water $NO_3^-$
concentration and organic carbon content in the sediment (Deek et al., 2013). In our study, the bottom water nitrate
concentration is too low (<0.5 to 4.5 µmol $L^{-1}$) to sustain the observed denitrification rates, and thus the major
nitrate source fueling the observed denitrification must be coupled nitrification-denitrification fueled by
mineralization of sedimentary organic material. This is reflected in a weak, but significant, correlation between
gross nitrification and denitrification rates ($r^2 = 0.35$). In our study, we find that this coupled nitrification-
denitrification has a strong influence on the total N flux. Denitrification accounts to 7.2 % (±1.3 %) of the total
supply of mineralized N (i.e., gross ammonification) at the impermeable sediment station NOAH-C, ~29.1 % (±0.9
%) at the semi-permeable sediment station NOAH-D and ~17.1 % (±2.2 %) at permeable sediment stations
(NOAH-A, NOAH-E). In permeable sediments, only a part of the freshly produced nitrate escaped to the water
column, whereas a large part was denitrified again in sediments. Denitrification removed 67 % of internally
produced $NO_3^-$ in permeable sediments, ~45 % in moderately permeable sediment and ~37 % in impermeable
sediment, respectively.
Our study covers diverse sediment types across the German Bight, but is based on core incubations and therefore
potentially underestimates advective processes. In a recent study by Neumann et al. (2017), the authors used $NO_3^-$
pore water profiles to calculate the $NO_3^-$ consumption rates across a similar range of North Sea sediments. They
extrapolated their nitrate consumption rates to the entire area of the German Bight based on a permeability
classification of sediments. They propose that ~24 % of sediments in the southern North Sea (German Bight) are
impermeable sediments (12,200 $km^{-2}$), ~39 % are moderately permeable sediments (19,600 $km^{-2}$) and ~37 %
(18,800 $km^{-2}$) are permeable sediments. They estimated that permeable sediment were the most efficient $NO_3^-$ sink
accounting for up to 90 % of the total benthic $NO_3^-$ consumption. In our assessment, which might better represents
the role of nitrification, we arrive at a somewhat lower contribution of ~80 % of total denitrification occurring in
moderately permeable and permeable sediments. Based solely on our data, we estimate a total nitrogen removal
of ~894 t N $d^{-1}$ in our study area. This daily $N_2$ production is close to the total N input (~1.000 t N $d^{-1}$) by the main
rivers Maas, Rhine, North-Sea Canal, Ems, Weser and Elbe (Pätsch and Lenhart, 2004), and, as such, underscores
the role of coastal sediments to counteract the eutrophication in the North Sea.
Our assessment, however, does not account for advective fluxes. Based on the same data set of permeability for
classification of different sediment types that Neumann et al (2017) used, we merge our dataset with the
assumptions of Neumann et al. (in preparation) to arrive at an improved estimate of sediment denitrification,



including nitrification as a source, but also accounting for the increasing importance of advection in permeable
sediments.
For impermeable sediments, advection can be neglected. The ratio of diffusive to adjective processes in moderately
permeable sediments is close to 1, which suggest that, if both processes act simultaneously, our diffusion-driven
estimate can be doubled. For permeable sediments, advection is far more important. Neumann et al. (2017) suggest
that advective fluxes exceed diffusive fluxes by a factor of up to 250. If this holds true, the measurements by far
underestimate N-cycling in permeable sediments. However, employing a factor of 250 to correct the observed
denitrification rates obviously exaggerates denitrification estimates, which even exceed the simultaneously
measured in-situ respiration rates of Ahmerkamp et al. (2015), indicating the limiting role of organic matter supply
(see below). A further reason for this overestimation is the fact that the solute transport in our core incubations
was not limited solely to molecular diffusion, but was substantially enhanced by faunal activity.
In the following, we aim to set an upper limit of denitrification based on primary production since denitrification
requires organic carbon, which is ultimately provided by pelagic primary production. For the freshwater influenced
regions of the German Bight, Capuzzo et al. (2018) assume a C fixation of 1.05 g C $m^{-2}$ $d^{-1}$. For an estimate of the
maximum denitrification rate we assume that 10 % of the fixed C is processed in the sediment (Heip et al., 1995)
and that all carbon is remineralized in the sediment by denitrification. Based on the stoichiometry of denitrification
(~12 g / mol C, ~14 g / mol N), this translates to [1.05 g * 10 % / 12 C * 14N =] 0.123 mg N that is removed per
$m^{-2}$ and day, or 9 mmol N $m^{-2}$ $d^{-1}$. This sets an absolute upper limit to the additional denitrification that could occur
in permeable sediments if all benthic C were remineralized by denitrification. Based on annual nitrate budgets,
Hydes et al. (1999) and van Beusekom et al. (1999) derived average denitrification rates of 0.7 mmol N $m^{-2}$ $d^{-1}$.
These rates, based on annual budgets, are somewhat lower than our incubation-based summer estimates in the
range of 1.1 to 1.4 mmol N $m^{-1}$ $d^{-1}$ (Tab. 1).
Seitzinger and Giblin (1996) linked benthic respiration and denitrification directly to the pelagic primary
production. By employing their formulas and using the primary production rates by Capuzzo et al. (2018), the
annual average of the sediment oxygen demand would be 14.3 mmol $O_2$ $m^{-2}$ $d^{-1}$ (1.05 g C $d^{-1}$ $m^{-2}$ = 87.5 mmol C
$d^{-1}$ $m^{-2}$), which corresponds to a benthic denitrification rate of 3.3 mmol N $m^{-2}$ $d^{-1}$. Since the annual average of
actually measured oxygen fluxes are close to this estimate (15.4 ± 12.9 mmol $O_2$ $m^{-2}$ $d^{-1}$, N=175) (Neumann et al.,
in preparation), we are confident that our denitrification estimates of up to 1.4 mmol N $m^{-2}$ $d^{-1}$ are reasonable.
**5 Summary and concluding remarks**
We evaluated a range of sedimentary nitrogen turnover pathways and found that ammonification in sediments is
an important N-source for primary production in the water column of the southeastern North Sea during summer.



Depending on water depth, 13-61 % of the estimated water column primary production is fueled by sedimentary
N release. Assimilation, and nitrification act as the main sinks of $NH_4^+$ mineralized from sedimentary organic
matter. Ultimately, the main factors governing nitrification are organic matter content / ammonification and
oxygen penetration depth in the sediment. The share of newly nitrified $NO_3^-$ reduced to $N_2$ amounts to two thirds
of $NO_3^-$ in permeable sediments, to nearly one half in moderately permeable sediment, and to one third in
impermeable sediments. We further showed that moderately permeable and permeable sediments account for up
to ~80 % of the total benthic $N_2$ production (~894 t N $d^{-1}$) in the southern North Sea (German Bight) during
summer, and neutralize nearly the total N input by main rivers (e.g. Elbe, Ems, Rhine, Weser) flowing into the
southern North Sea (~1.000 t N $d^{-1}$). Thus impermeable sediments act as an important N source for primary
producers, whereas moderately permeable and permeable sediments comprise a main reactive N sink counteracting
eutrophication in the North Sea. Seasonal and spatial variabilities, especially from nearshore to offshore, should
be evaluated in future studies.
**Acknowledgements**
We thank the captain and the crew of *R/V Heincke* for their support during the sampling campaigns. M. Birkicht
from the Leibniz Centre for Tropical Marine Research (ZMT) in Bremen is gratefully acknowledged for his
assistance with nutrient measurements. We further thank E. Logemann for the analysis of macrobenthos.

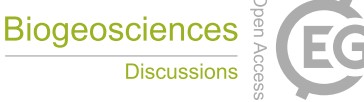

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

Contributions of Heterotrophic Denitrification and Anammox to Nitrogen Removal in the OMZ Waters of the
Ocean, Methods in Enzymology, 486, 223-251, 2011.
Hydes, D. J., Kelly-Gerreyn, B. A., Le Gall, A. C., and Proctor, R.: The balance of supply of nutrients and
demands of biological production and denitrification in a temperate latitude shelf sea – a treatment of the
southern North Sea as an extended estuary, Marine Chemistry, 68, 117-131, 1999.
Jensen, K. M., Jensen, M. H., and Kristensen, E.: Nitrification and denitrification in Wadden Sea sediments
(Konigshafen, Island of Sylt, Germany) as measured by nitrogen isotope pairing and isotope dilution, 11, 181-
485  191, 1996.
Jorgensen, B. B.: Mineralization of organic matter in the sea bed-the role of sulphate reduction, Nature, 296,
487  643-645, 1982.
Jorgensen, B. B.: Processes at the sediment-water interface. In: The Major Biogeochemical Cycles and Their
Interactions, Bolin, B. and Cook, R. B. (Eds.), John Wiley, New York, 1983.
Jorgensen, B. B., Bang, M., and Blackburn, T. H.: Anaerobic mineralization in marine sediments from the Baltic
Sea-North Sea transition, Marine Ecology Progress Series, 59, 39-54, 1990.
Kadlec, R. H. and Wallace, S. D.: Treatment Wetlands, Taylor & Francis Group, Boca Raton
London, New York, 2009.
Kana, T. M., Darkangelo, C., Hunt, M. D., Oldham, J. B., Bennett, G. E., and Cornwell, J. C.: Membrane Inlet
Mass Spectrometer for Rapid High-Precision Determination of $N_2$, $O_2$, and Ar in Environmental Water
Samples.pdf>, Analytical Chemistry, 66, 4166-4170, 1994.
Kérouel, R. and Aminot, A.: Fluorimethic determination of ammonia in  sea and estuarine water by direct
segmented flow analysis. , Marine Chemistry, 57, 265-275, 1997.
Koike, I. and Hattori, A.: Simultaneous determinations of nitrification and nitrate reduction in coastal sediments
by a $^{15}$N dilution technique, Appl Environ Microbiol, 35, 853-857, 1978.
Krämer, K., Holler, P., Herbst, G., Bratek, A., Ahmerkamp, S., Neumann, A., Bartholoma, A., van Beusekom, J.
E. E., Holtappels, M., and Winter, C.: Abrupt emergence of a large pockmark field in the German Bight,
southeastern North Sea, Sci Rep, 7, 5150, 2017.
Kristensen, E., Andersen, F. O., Holmboe, N., Holmer, M., and Thongtham, N.: Carbon and nitrogen
mineralization in sediments of the Bangrong mangrove area, Phuket, Thailand, Aquatic Microbial Ecology, 22,
506  199-213, 2000.
Lenhart, H. J. and Pohlmann, T.: The ICES-boxes approach in relation to results of a North Sea circulation
model, Tellus A: Dynamic Meteorology and Oceanography, 49, 139-160, 1997.
Lohse, L., Malschaert, J. F. P., Slomp, C. P., Helder, W., and van Raaphorst, W.: Nitrogen cycling in the North
Sea sediments: interaction of denitrification and nitrification of offshore and coastal areas, Marine Ecology
Progress Series, 101, 283-296, 1993.
LOICZ: Land-Ocean Interactions in the Coastal Zone, 1995.
Los, F. J., Troost, T. A., and Van Beek, J. K. L.: Finding the optimal reduction to meet all targets—Applying
Linear Programming with a nutrient tracer model of the North Sea, Journal of Marine Systems, 131, 91-101,
515  2014.
Mackin, J. E. and Swider, K. T.: Organic matter decomposition pathways and oxygen consumption in coastal
marine sediments, Journal of Marine Research, 47, 681-716, 1989.
Marchant, H.: Nitrogen cycling in coastal permeable sediments from euthrophied regions, PhD, Fachbereich
Geowissenschaften, Universität Bremen, Bremen, 1-274 pp., 2014.
Marchant, H. K., Holtappels, M., Lavik, G., Ahmerkamp, S., Winter, C., and Kuypers, M. M. M.: Coupled
nitrification-denitrification leads to extensive N loss in subtidal permeable sediments, Limnology and
Oceanography, 61, 1033-1048, 2016.
Meysman, F. J., Middelburg, J. J., and Heip, C. H. R.: Bioturbation: a fresh look at Darwin's last idea, Trends
Ecology Evolution, 21, 688-695, 2006.
Miyajima, T., Wada, E., Hanba, Y. T., and Vijarnsorn: Anaerobic mineralization of indigenous organic matters
and methanogenesis in tropical wetland soils, Geochimica et Cosmochimica Acta, 61, 3739-3751, 1997.
Myers, R. J. K.: Temperature effects on ammonificaiton and nitrification in a tropical soil, Soil Biology and
Biochemistry, 7, 83-86, 1975.



Neubacher, E. C., Parker, R. E., and Trimmer, M.: The potential effect of sustained hypoxia on nitrogen cycling
in sediment from the southern North Sea: a mesocosm experiment, Biogeochemistry, 113, 69-84, 2012.
Neubacher, E. C., Parker, R. E., and Trimmer, M.: Short-term hypoxia alters the balance of the nitrogen cycle in
coastal sediments, Limnology and Oceanography, 56, 651-665, 2011.
Neumann, A.: Elimination of reactive nitrogen in continental shelf sediments measured by membrane inlet mass
spectrometry., PhD, Department Geowissenschaften, Universität Hamburg, Hamburg, 2012.
Neumann, A., van Beusekom, J. E. E., Eisele, A., Emeis, K.-C., Friedrich, J., Kröncke, I., Logemann, E. L.,
Meyer, J., Naderipour, C., Schückel, U., Wrede, A., and Zettler, M.: Elucidating the impact of macrozoobenthos
on the seasonal and spatial variability of benthic fluxes of nutrients and oxygen in the southern North Sea, in
preparation. in preparation.
Neumann, A., van Beusekom, J. E. E., Holtappels, M., and Emeis, K.-C.: Nitrate consumption in sediments of
the German Bight (North Sea), Journal of Sea Research, 127, 26-35, 2017.
Nichols, F. H. and Thompson, J. K.: Time scales of change in the San Francisco Bay benthos, Hydrobiologia,
542    129, 121-138, 1985.
Nishio, B. L., Komada, M., Arao, T., and Kanamori, T.: Simultaneous determination of transformation rates of
nitrate in soi, Japan Agricultural Research Quarterly: JARQ, 35, 11-17, 2001a.
Nishio, T., Komada, M., Arao, T., and Kanamori, T.: Simultaneous determination of transformation rates of
nitrate in soil, Japan Agricultural Research Quarterly: JARQ, 35, 11-17, 2001b.
OSPAR: Quality Status Report, London, 176 pp pp., 2010.
Pätsch, J. and Lenhart, H.-J.: Daily loads of nutrients, total alkalinity, dissolved inorganic carbon and dissolved
organic carbon of the European continental rivers for the years 1977-2002. In: Berichte aus dem Zentrum für
Meeres- und Klimaforschung, Reihe B: Ozeanographie, University of Hamburg, Germany, 2004.
Redfield, A. C.: The biological control of chemical factors in the environment, American Scientist, 46, 205-221,
552    1958.
Reise, K. and Kohlus, J.: Seagrass recovery in the Northern Wadden Sea?, Helgoland Marine Research, 62, 77-
554    84, 2007.
Reise, K. and Siebert, I.: Mass occurrence of green algae in the German Wadden Sea, Deutsche Hydrographische
Zeitschrift, 1, 171-188, 1994.
Rick, H. J., Rick, S., Tillmann, U., Brockmann, U., Gärtner, U., Dürselen, C., and Sündermann, J.: Primary
Productivity in the German Bight (1994–1996), Estuaries and Coasts, 29, 4-23, 2006.
Rysgaard, S., Risgaard-Petersen, N., and Sloth, N. P.: Nitrification, denitrification and nitrate ammonification in
two coastal lagoons in Southern France, Hydrobiologia, 329, 133-141, 1996.
Sanders, T., Schöl, A., and Dähnke, K.: Hot spots of nitrification in the Elbe Estuary and their impact on nitrate
regeneration, Estuaries and Coasts, 41, 128-138, 2018.
Seitzinger, S. P. and Giblin, A. E.: Estimating denitrification in North Atlantic continental shelf sediments,
Biogeochemistry, 35, 235-260, 1996.
Sigman, D. M., Casciotti, K. L., Andreani, M., Barford, C., Galanter, M., and Böhlke, J. K.: A bacterial method
for the nitrogen isotopic analysis of nitrate in seawater and freshwater, Anal. Chem., 73, 4145-4153, 2001.
Sumi, T. and Koike, I.: Estimation of ammonification and ammonium assimilation in surfkial coastal and
estuarine sediments, Limnology and Oceanography, 35, 270-286, 1990.
Thibodeau, B., Lehmann, M. F., Kowarzyk, J., Mucci, A., Gélinas, Y., Gilbert, D., Maranger, R., and Alkhatib,
M.: Benthic nutrient fluxes along the Laurentian Channel: Impacts on the N budget of the St. Lawrence marine
system, Estuarine, Coastal and Shelf Science, 90, 195-205, 2010.
van Amstel, M., de Neve, W., de Kraker, J., and Glasbergen, P.: Assessment of the potential of ecolabels to
promote agrobiodiversity, Ambio, 36, 551-558, 2007.
Van Beusekom, J., Brockmann, U. H., Hesse, K.-J., Hickel, W., Poremba, K., and Tillmann, U.: The importance
of sediments in the transformation and turnover of nutrients and organic matter in the Wadden Sea and German
Bight, German Journal of Hydrography, 51, 245-266, 1999.
van Leeuwen, S. M., van der Molen, J., Ruardij, P., Fernand, L., and Jickells, T.: Modelling the contribution of
deep chlorophyll maxima to annual primary production in the North Sea, Biogeochemistry, 113, 137-152, 2013.
von Westernhagen, H., Hickel, W., Bauerfeind, E., Niermann, U., and Kröncke, I.: Sources and effects of
oxygen deficiencies in the south-eastern North Sea, Ophelia, 26, 457-473, 1986.
Vouvé, F., Guiraud, G., Marol, C., Girard, M., Richard, P., and Laima, M. J. C.: $NH_4^+$ turnover in intertidal
sediments of Marennes-Oléron Bay (France): effect of sediment temperature, Oceanologica Acta, 23, 575-584,
583    2000.
Vymazal, J.: Constructed Wetlands for Wastewater Treatment, Water, 2, 530-549, 2010.
Vymazal, J.: Removal of nutrients in various types of constructed wetlands, Sci Total Environ, 380, 48-65, 2007.
Wankel, S. D., Mosier, A. C., Hansel, C. M., Paytan, A., and Francis, C. A.: Spatial variability in nitrification
rates and ammonia-oxidizing microbial communities in the agriculturally impacted Elkhorn Slough estuary,
California, Appl Environ Microbiol, 77, 269-280, 2011.



Ward, B. B.: Nitrification in Marine Systems. In: Nitrogen in the marine environment, Capone, D. G., Bronk, D.
A., Mulholland, M. R., and Carpenter, E. J. (Eds.), Academic Press, Burlington, Amsterdam, San Diego,
London, 2008.
Weston, K., Fernand, L., Mills, D. K., Delahunty, R., and Brown, J.: Primary production in the deep chlorophyll
maximum of the central North Sea, Journal of Plankton Research, 27, 909-922, 2005.
Zhang, L., Altabet, M. A., Wu, T., and Hadas, O.: Sensitive Measurement of $NH_4^+$ $^{15}N/^{14}N$ ($\delta^{15}NH_4^+$) at Natural
Abundances Levels in Fresh and Saltwaters, Anal Chem, 79, 5297-5303, 2007.





**Table 1: Rates of nitrification, dissimilatory nitrogen reduction to ammonia (DNRA), anaerobic ammonia oxidation**
**(anammox) and denitrification (DNIT) (in μmol N m$^{-2}$ d$^{-1}$) in the North Sea of other published data. Abbreviation of**
**methods: SIDM - sediment isotope dilution method; MABT - modified acetylene block technique; SSI - sediment slurry**
**incubations, PWMI – pore-water mean fitting, IPT - isotope-pairing technique.**

| Location | Nitrification | DNRA | Anammox | DNIT rate / NO$_3^-$ uptake | Sediment type | C$_{org}$ | C:N | Sampling time | Method | Reference |
|---|---|---|---|---|---|---|---|---|---|---|
| | [μmol m$^{-2}$ d$^{-1}$] | | | | [-] | [% dry wt] | [atom] | | [-] | |
| German Bight (North Sea) | 728 ±444 | N.D. | N.D. | 1095 ±596* | medium sand | 0.03 | <0.01 | 08./09.2016 | SIDM | this study |
| | 1,090 ±312 | N.D. | N.D. | 1371 ±850* | | 0.04 | 0.01 | | | |
| | 1,493 ±211 | N.D. | N.D. | 1350 ±982* | Fine sand | 0.21 | 0.03 | | | |
| | 1,233 ±978 | N.D. | N.D. | 1198 ±427* | clay/silt | 0.73 | 0.10 | | | |
| Dutch Coast | N.D. | N.D. | 0.0 | N.D. | fine sand | 0.03 | N.D. | 11.2010 | SSI | Bale et al., 2014 |
| | | | 0.2 | | | | | 02.2011 | | |
| | | | 1.3 | | | | | 05.2011 | | |
| | | | 0.6 | | | | | 08.2011 | | |
| Oyster Ground | N.D. | N.D. | 0.0 | N.D. | muddy sand / clay / silt | 0.30 | N.D. | 11.2010 | | |
| | | | 2.3 | | | | | 02.2011 | | |
| | | | 10.4 | | | | | 05.2011 | | |
| | | | 12.8 | | | | | 08.2011 | | |
| North Dogger | N.D. | N.D. | 0.0 | N.D. | fine sand | 0.03 | N.D. | 11.2010 | | |
| | | | 0.8 | | | | | 02.2011 | | |
| | | | 0.0 | | | | | 05.2011 | | |
| | | | 1.1 | | | | | 08.2011 | | |
| Elbe Estuary / coastal zones | N.D. | N.D. | N.D. | 771* | coarse sand | 0.6 | 6.0 | 03.2009 | IPT | Deek et al., 2013 |
| | | | | 1215* | | 0.1 | N.D. | | | |
| | | | | 3200* | | 0.1 | | | | |
| | | | | 864* | | 0.6 | 6.0 | | | |
| | | | | 1425* | | 0.2 | | 09.2009 | | |
| | | | | 47* | | 0.1 | N.D. | | | |
| | | | | 140* | | 0.2 | | | | |
| Oyster Ground | 288 ±144 | N.D. | N.D. | 12.0* | muddy sand | 0.12 | 6.0 | 08.1991 | MABT | Lohse et al., 1993 |
| | 192 ±96 | | | 19.2* | | | | 02.1992 | | |
| Weiss Bank | 216 | | | 21.6* | | 0.16 | 8.0 | 08.1991 | | |
| | 120 ±120 | | | 16.8* | | | | 02.1992 | | |
| Tail End | 432 ±168 | | | 2.4* | fine sand | 0.16 | 5.3 | 08.1991 | | |
| | 264 ±120 | | | 0* | | | | 02.1992 | | |
| Esbjiberg | 408 ±216 | | | 9.6* | | 0.06 | 6.0 | 08.1991 | | |
| | 168 ±168 | | | 91.2* | | | | 02.1992 | | |
| Helgoland | 0 | | | 45.6* | silt | 1.28 | 8.5 | 08.1991 | | |
| | 216 ±1220 | | | 196.8* | | | | 02.1992 | | |
| Elbe Rinne | 264 ±72 | | | 4.8* | muddy sand | 0.46 | 9.2 | 08.1991 | | |
| | 288 ±96 | | | 31.2* | | | | 02.1992 | | |
| Frisian Front | 624 ±288 | | | 16.8* | | 0.46 | 9.2 | 08.1991 | | |
| | 192 ±72 | | | 24.0* | | | | 02.1992 | | |
| Sylt | 81.6 ±64.8 | N.D. | N.D. | 372 ±132* | coarse sand | N.D. | N.D. | 06.1993 | IPT, SIDM | Jensen et al., 1996 |
| | 11 ±2 | | | 44.5 ±13.5* | | | | 04.1994 | | |
| | 3.8 ±1.6 | | | 17 ±4* | fine sand | | | 04.1994 | | |
| | 1,116 ±924 | | | 75 ±39* | muddy sand | | | 03.1993 | | |
| | 19.5 ±9.5 | | | 103.5 ±17.5* | | | | 04.1994 | | |
| Helgoland | 1,150 ±700 | 20 ±5 | N.D. | 870 ±100* | fine sand | N.D. | N.D. | 05.2012 | SIDM | Marchant et al., 2016 |
| | 210 ±50 | 250 ±50 | | 2,280 ±300* | medium sand | | | | | |
| | 2,980 ±420 | 110 ±60 | | 520 ±30* | coarse sand | | | | | |
| Sean Gras | N.D. | N.D. | 24.0 | 48* | medium sand | 0.05 | 8.1 | 04.2007 | IPT | Neubacher et al., 2011 |
| | | | 24.0 | 72* | | 0.06 | 7.4 | 05.2007 | | |
| | | | 0 | 120* | | 0.10 | 8.5 | 09.2007 | | |
| | | | 48.0 | 144* | | 0.05 | 6.6 | 10.2007 | | |
| | | | 0 | 24* | | N.D. | N.D. | 04.2008 | | |
| Oyster Ground | N.D. | N.D. | 24 | 288* | muddy sand | 0.28 | 10.2 | 02.2007 | | |
| | | | 24 | 120* | | 0.22 | 9.0 | 04.2007 | | |
| | | | 24 | 120* | | 0.20 | 8.4 | 05.2007 | | |
| | | | 120 | 408* | | 0.22 | 9.2 | 09.2007 | | |
| | | | 144 | 504* | | 0.23 | 9.4 | 10.2007 | | |
| | | | 48 | 144* | | 0.27 | 8.7 | 04.2008 | | |
| North Dogger | | | 0 | 24* | muddy sand | 0.45 | 10.2 | 02.2007 | | |
| | | | 0 | 96* | | 0.45 | 9.4 | 04.2007 | | |
| | | | 24 | 168* | | 0.42 | 9.7 | 05.2007 | | |
| | | | 48 | 288* | | 0.46 | 9.7 | 09.2007 | | |
| | | | 48 | 264* | | 0.38 | 9.6 | 10.2007 | | |
| German Bight / Dogger Bank | N.D. | N.D. | N.D. | 20.5 ±4.5** | mud | 0.37 ±0.02 | N.D. | 05.2009 | PWMF | Neumann et al., 2017 |
| | | | | 28.5 ±23.5** | | 0.16 ±0.12 | | 02.2010 | | |
| | | | | 8 ±8** | muddy sand | 0.13 ±0.10 | | 05.2009 | | |
| | | | | 12.5 ±12.5** | | 0.10 ±0.08 | | 02.2010 | | |
| | | | | 59.5 ±25.5** | sand | 0.16 ±0.13 | | 05.2009 | | |
| | | | | 99 ±35.0** | | 0.02 | | 02.2010 | | |

N.D. – not determined
* Denitrification
** NO$_3^-$ uptake





**Table 2: Characteristics of bottom water and sediment characteristics of the sampled stations in the North Sea (https://doi.org/10.1594/PANGAEA.846041). $C_{org}$ means organic carbon content and TN means total nitrogen content of the surface sediment.**

| Location | Depth | Sediment core / Chamber | Method | Sediment type | $C_{org}$ | TN | Porosity | Permea-bility | Temp. | Salinity | OPD | Bottom water concentration | | |
|---|---|---|---|---|---|---|---|---|---|---|---|---|---|---|
| | | | | | | | | | | | | $NH_4^+$ | $NO_2^-$ | $NO_3^-$ |
| [-] | [m] | | [-] | | [%] | | [-] | [m²] | [°C] | [PSU] | [mm] | µmol L⁻¹ | | |
| NOAH-A | 31.0 | 1 | ex-situ | medium sand | 0.03* | ≤0.01* | 0.37 | 1.7*10⁻¹⁰ | 19.1 | 33.7 | >15 | 0.5 | 0.4 | 0.9 |
| | | 2 | | | | | | | | | | 1.5 | 0.1 | 0.8 |
| | | 3 | | | | | | | | | | 1.8 | 0.0 | 2.4 |
| | | 4 | | | | | | | | | | 1.2 | 0.1 | 1.9 |
| NOAH-C | 25.4 | 1 | ex-situ | clay/silt | 0.73 | 0.10 | 0.56 | 1*10⁻¹⁵ | 19.1 | 32.5 | 3.6 | 4.3 | 0.1 | 1.9 |
| | | 2 | | | | | | | | | | 2.3 | 0.1 | 1.4 |
| | | 3 | | | | | | | | | | 7.1 | 0.2 | 2.9 |
| NOAH-D | 38.0 | 1 | ex-situ | fine sand | 0.21 | 0.03 | 0.43 | 1.4*10⁻¹³ | 18.9 | 33.0 | 2.4 | 2.2 | 0.1 | 1.2 |
| | | 2 | | | | | | | | | | 1.7 | 0.0 | 0.7 |
| | | 3 | | | | | | | | | | 2.5 | 0.1 | 0.6 |
| NOAH-E | 28.4 | 1 | ex-situ | medium sand | 0.04 | 0.01 | 0.29 | 8.8*10⁻¹² | 18.7 | 32.4 | 4.2 | 3.3 | 0.5 | 2.1 |
| | | 2 | | | | | | | | | | 9.6 | 0.7 | 1.0 |
| | | 3 | | | | | | | | | | 3.6 | 1.5 | 4.5 |
| | | 4 | | | | | | | | | | 2.9 | 1.2 | <0.5 |

\* estimated

**Table 3: Rates of $NH_4^+$ assimilation, benthic net $NO_3^-$ and benthic net $NH_4^+$ fluxes per area, water depth below thermocline and concentration of dissolved inorganic nitrogen (DIN) in the thermocline. Bottom water concentration of nitrate ($cNO_3^-$), nitrite ($cNO_2^-$) and ammonium ($cNH_4^+$). The concentration of DIN per area was calculated by the multiplication of the water depth below the thermocline with the concentration of DIN. Turnover rates of nitrogen were calculated by the division of DIN per area with the rates of $NH_4^+_{net}$, $NO_3^-_{net}$ and $NH_4^+_{ass}$ and the effect of sedimentary N release on the reactive nitrogen available for primary production in the water column.**

| Station | $rNH_4^+_{net} + rNO_3^-_{net} + rNH_4^+_{ass}$ | Water depth below thermocline | $cNO_3^-$ | $cNO_2^-$ | $cNH_4^+$ | DIN per area | N turnover | sedimentary N support for primary production |
|---|---|---|---|---|---|---|---|---|
| [-] | [mmol m⁻² d⁻¹] | [m] | [µmol L⁻¹] | | | [mmol m⁻²] | [days] | [%] |
| NOAH-A | 2.0 ±0.6 | 29.5 | 0.1 | <0.1 | 0.6 ±0.2 | 20.7 | 0.7 | 17.3 |
| NOAH-C | 7.7 ±3.0 | 10.0 | <0.1 | 0.7 | 2.0 ±0.2 | 30.0 | 3.0 | 61.2 |
| NOAH-D | 1.2 ±0.1 | 38.0 | 0.1 ±0.1 | 0.1 | 0.8 ±0.6 | 26.6 | 0.7 | 12.8 |
| NOAH-E | 4.1 ±0.9 | 10.0 | <0.1 | <0.1 | 0.3 ±0.1 | 3.0 | 0.7 | 35.2 |

**Table 4: Sediment permeability classes with the area in the German Bight and rates of $NO_3^-$ consumption and $N_2$ production in the sediment. Estimated $NO_3^-$ consumption rates from Neumann et al. (2017).**

| Sediment type | Area | $NO_3^-$ consumption | $N_2$ production |
|---|---|---|---|
| [-] | [km²] | [mol d⁻¹] | |
| Impermeable $k < 3*10^{-13}$ m² | 12,200 | 0.4*10⁷ ±0.2*10⁷ | 0.7*10⁷ ±0.3*10⁷ |
| Mod. permeable | 19,600 | 1.4*10⁷ ±0.2*10⁷ | 1.3*10⁷ ±1.0*10⁷ |
| Permeable $k > 3*10^{-12}$ m² | 18,800 | 0.6*10⁷ ±0.4*10⁷ | 1.1*10⁷ ±0.6*10⁷ |
| Weighted average total | 50,600 | 2.0*10⁷ ±0.8*10⁷ | 3.1*10⁷ ±2.0*10⁷ |





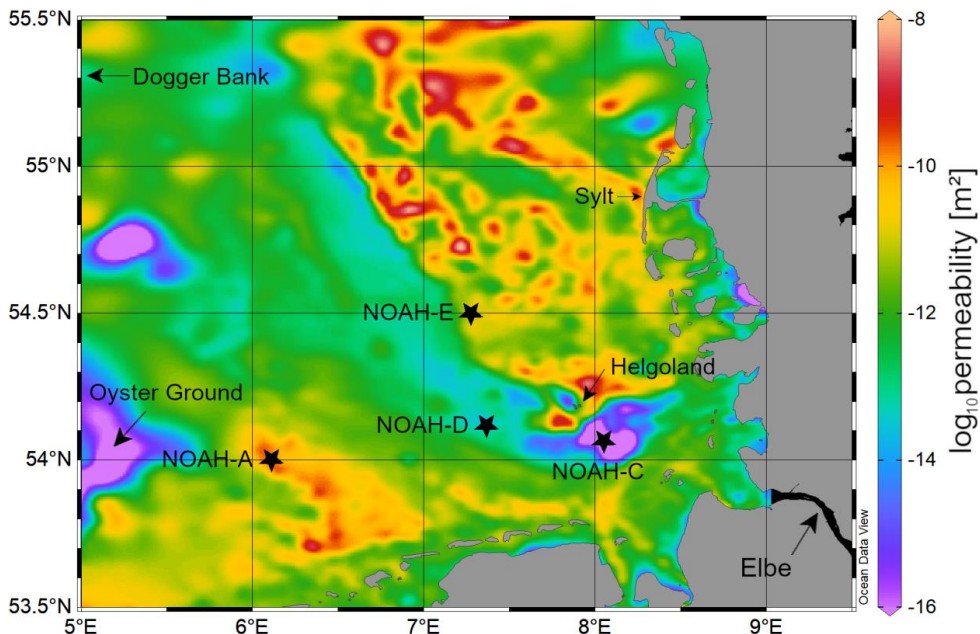

**Figure 1: Map showing the sampling stations NOAH-A, NOAH-C, NOAH-D and NOAH-E in the German Bight in the North Sea. Colored areas show the spatial variability of surface sediment permeability (https://doi.org/10.1594/PANGAEA.872712).**

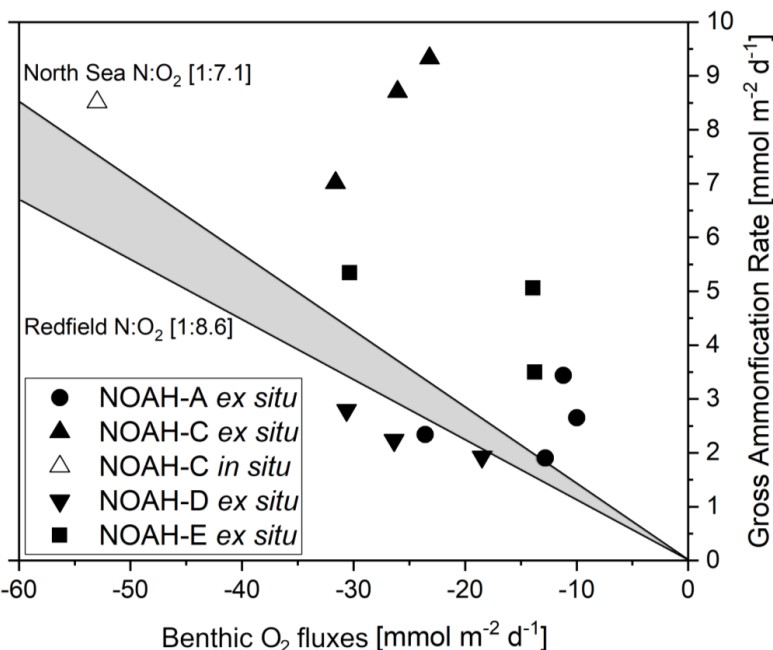

**Figure 2: Benthic O₂ fluxes and gross ammonification rates of the sampled stations. The lines show the Redfield ratio of oxygen and nitrogen (N:O₂ 1:8.625) (Redfield, 1958) and of the oxygen and nitrogen ratio determined by the C/N ratio in the North Sea (N:O₂ 1:7.1).**



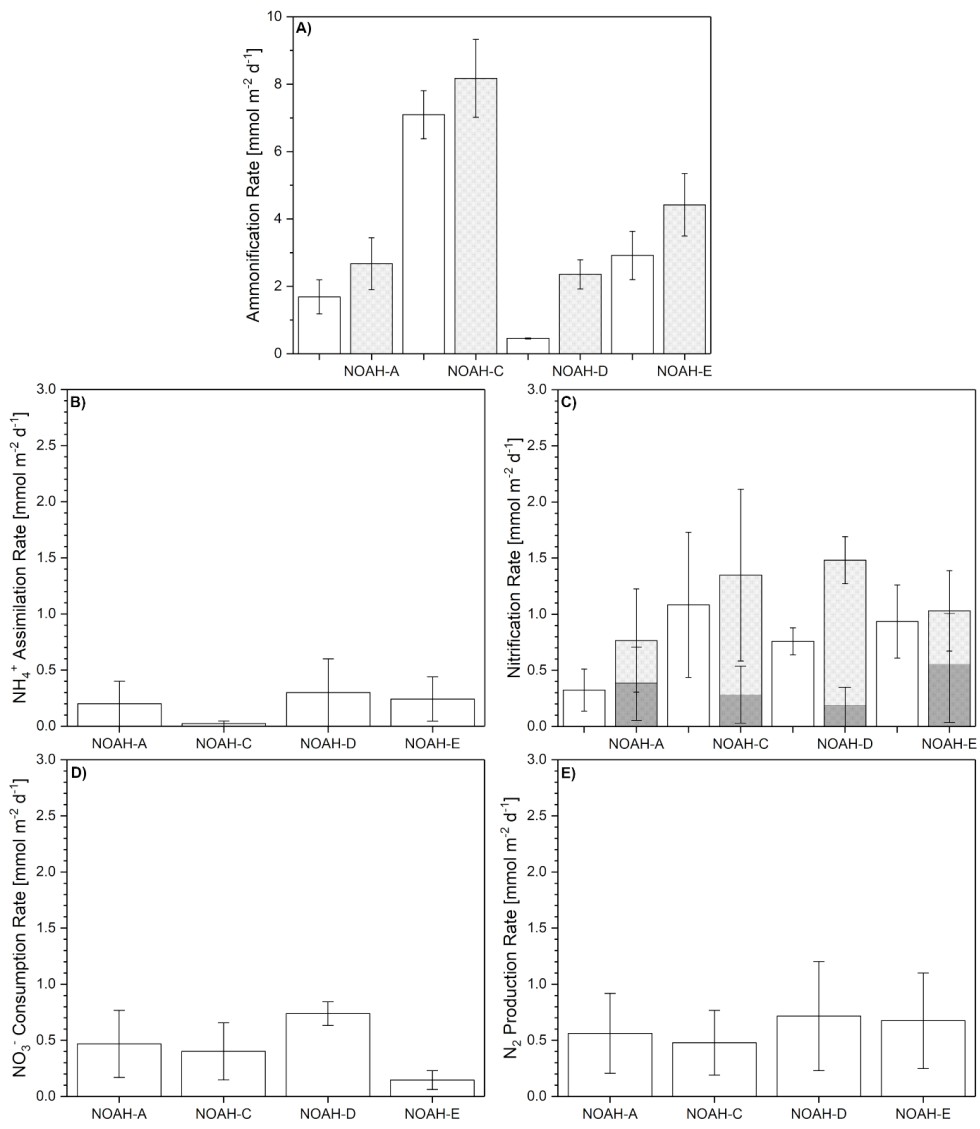

**Figure 3: Benthic N-transformation rates (in mmol m⁻² d⁻¹) of gross (grey) and net (white) ammonification (A), assimilation (B), nitrification, where white bars are net nitrification, light grey colored bars are complete gross nitrification (bottom water and sediment) and dark grey colored bars show sedimentary nitrification (C), nitrate consumption (D) and N₂ production rates (E) of the stations NOAH-A (permeable sediment), NOAH-C (impermeable sediment), NOAH-D (moderately permeable sediment) and NOAH-E (permeable sediment).**


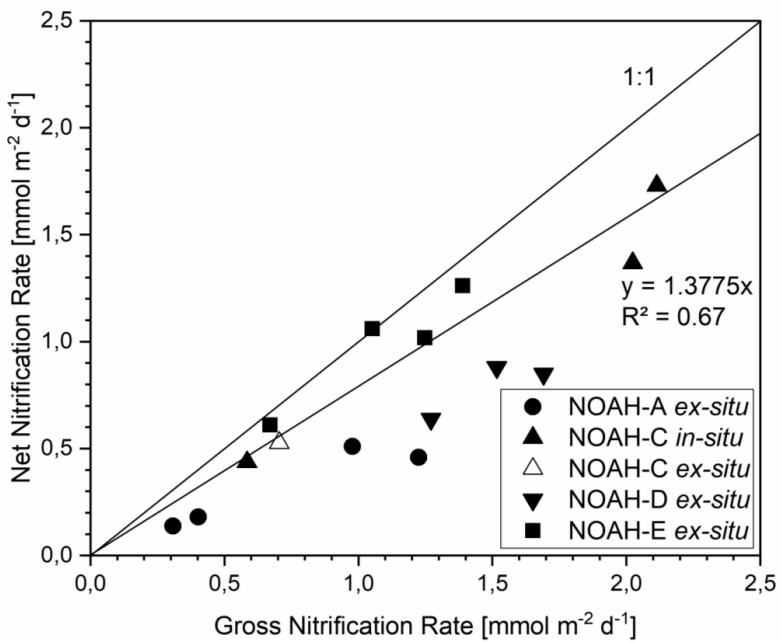

**Figure 4: Correlation of gross and net nitrification rates. The lines shows the 1:1 ratio and the slope of the samples.**
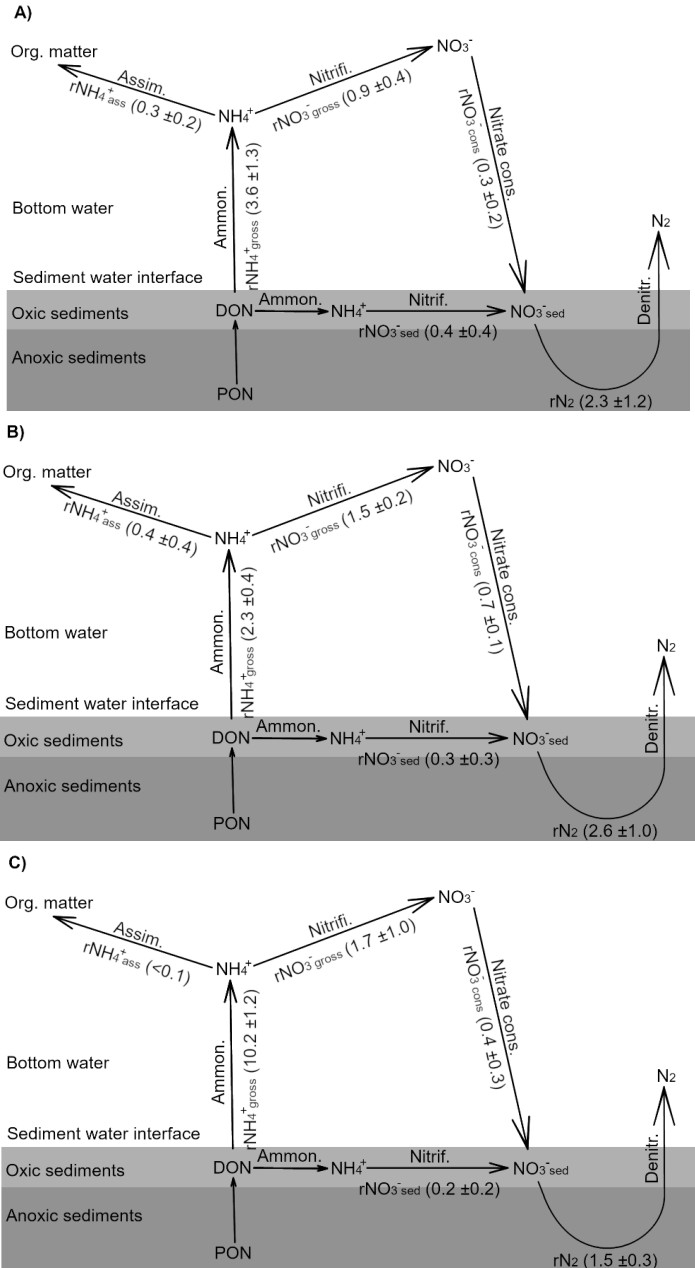

**Figure 5: Benthic N-transformation rates of ammonification (ammon.), assimilation (assim.), nitrification (nitri.), nitrate consumption (nitrate cons.) and $N_2$ production rates (denitr.) in the southern North Sea (German Bight) in mol N $d^{-1}$. They values given in (A), (B) and (C) multiply with $10^7$. PON means particulate organic nitrogen and DON is dissolved organic nitrogen. (A) shows the N-transformation rates in permeable sediments ($k > 3*10^{-12}$ $m^2$), (B) in moderately permeable sediments ($k = 3*10^{-12}$ to $3*10^{-13}$ $m^2$) and (C) in impermeable ($k < 3*10^{-13}$ $m^2$) sediments.**