# Peer review of "Spatial variations in sedimentary N-transformation rates in the North Sea (German Bight)"

_Biogeosciences, 2019_

## Referee Comment (RC1) · Anonymous Referee #1 · 28 Aug 2019

General appreciation

Interesting manuscript, generally well written but lacking in clarity at several points. This makes the conclusions stretched in view of the inability of the reader to properly assess what was done, and how. Perhaps a bit rushed in several instances (missing information, repeated words, misplaced words, reference list seemingly too long compared to actual citations in text, etc). Some important statements are inadequately substantiated. I would suggest a careful re-write, addressing the main concerns below – interpretation of data, discussion (including scale-up to whole basin) and conclusions seem unassailable without this being done.

Main concerns

- Lines 71-74. It is suggested that the isotope dilution method employed in this study can 'unravel several N-processes like ammonification, assimilation, nitrification, denitrification (. . .)', DNRA. . .etc within sediments. This requires demonstration, which is not clearly apparent either after this statement or in fact within the rest of the manuscript. The authors seem to use 15N isotope dilution methods as first developed by Koike and Hattori (1977). This is a powerful technique, but not without issues. These have been well documented by the Aarhus group in the 80's and 90's. The main issue is the underestimation of both nitrification and denitrification due to the inability of the technique employing only $15NO_3-$ to adequately account for coupled nitrification-denitrification, even if this is paired with other methods – such as the acetylene inhibition technique or any kind of budget modelling (as in fact supports the study cited by the authors as Nishio et al 2001b). The solution attempted in some cases (Nishio, et al., 1983) passes through using the $15NH_4+$ dilution technique of Blackburn (1979) in parallel, requiring the simultaneous incubation of two core samples – one amended with isotopically labelled nitrate and the other ammonium. Denitrification could be then estimated as the sum of denitrification rates measured in the two cores, and both nitrification and coupled nitrification-denitrification from the $15NH_4+$ labelled core. Indeed, it seems that was what was done (Lines 113-115), with two main consequences: one, affecting the overall evaluation of uncertainty of the results of the various rates on evidence in this manuscript because fewer replicates of each incubation were available in reality (2 of each label for each station, instead of 4 mentioned in lines 107-109, and in the case of station D, only one, Line 113), making the account of propagation of uncertainty challenging (I will return to this point later); and two, an added uncertainty introduced by the amount of $15NH_4+$ added to the cores (as the authors state, $NH_4+$ concentration in bottom waters are usually very low, Lines 234-235) and how this would affect the accuracy of estimates of nitrification and coupled nitrification-denitrification (see for example, Henriksen & Kemp, 1988 and Rysgaard et al., 1993), particularly because these cores were quite bioturbated (Lines 256-260). I wouldn't find it surprising under these circumstances that it is difficult to explain the relatively high ammonification

rates compared to the benthic oxygen consumption (page 9) and the lack of correlation between gross ammonificatin and nitrification rates (Line 317).

Based on this general assessment, I would suggest the following issues addressed:

- Present a clearer description of the methodology used to unravel all the process rates mentioned. Specifically, the differences between net and gross rates, and the process by which sediment-water fluxes measured in core incubations are translated into process rates, perhaps with a diagram of the steps taken to discriminate between different rates. It should also be explained that the sediment-water exchanges measuring method would imply that these are steady state fluxes being measured. Are the fluxes measured in the same cores that were amended by the isotope labelled compounds? Etc – clarity on this front would help clarify some of the potential queries rising from reading the manuscript in its current form.

- Propagation of error and significance of differences encountered between different rates at different locations needs to be discussed appropriately, and concrete results presented in this regard. Rates are presented without appropriate attention to this point – and often calculated on the basis of sums or differences between terms with significant individual uncertainty (see above for total denitrification, which would be estimated from the sum of two rates obtained from two different core incubations, of which there is only one replicate in 75% of the cases and none for 25%). For example, in Table 3, in the second column, the sum of three process rates is presented with relatively low uncertainty – is this from a flux measurement and therefore this is an aggregate? (see above). Another example: see formula 3 (page 6). In NOAH -C (Line 201) to calculate assimilation, the difference between gross and net ammonification alone if propagation of error is accounted for would already be a highly uncertain number: ($8.3\pm2.3$ - $6.8\pm2.3$ = $1.5 \pm 3.25$) and we haven't yet subtracted the gross nitrification rate. The chosen mode of representing uncertainty is also not explained – how is it calculated (looking at the size of the error bars and the spread of results, I presume as the standard error of measurements, with n=2 or 1? Or n=4 or 3?), but has to be made explicit

– are we dealing with the standard error ($\sqrt{\sigma}$/n), $\sigma$, 2*$\sigma$, 3*$\sigma$? This step is in my view critical to properly asses the results in context.

- Present an explicit discussion of the potential pitfalls of using the 15N dilution technique without isotope pairing (Rysgaard, Nielsen et al). In particular, the issues with underestimation of denitrification and nitrification associated with the coupling of both processes and how they could affect interpretation of the results, not mentioned on the manuscript.

Additional points to address

Line 49-50: "very little is known about N cycling and N transformation rates in the sediment". This is a very strong statement and has to be amply justified although I see no evidence of such being the case in the literature. At least, it should be contextually set in a much better way. The justification coming afterwards is rather confusing – are the authors suggesting nothing is known about benthic N cycling in the North Sea because previous studies failed to distinguish between mineralization and nitrification?

---

## Referee Comment (RC2) · Anonymous Referee #2 · 11 Oct 2019

This is a well-written study of the sedimentary N cycling in the Southern North Sea. The novelty of this work is the quantification of ammonification rates and comparison of the nitrogen conversions in the different sediment types. Rates were determined with an isotope dilution method applied in laboratory incubations.

Key results include sedimentary ammonification and benthic N2 production rates. The concentrations and rates determined in the study provide valuable data for the assessment of the nitrogen conversions in the North Sea, and the role of the sediments as sinks, sources and sites of nitrogen regeneration and N release from the system. The study contributes baseline data that can be used in future studies for the evaluation of potential changes in the North Sea N-cycle.

My two main concerns with the study are related to the methods used. The first concern

is that the conclusions of the work are based on one sampling in summer 2016 (a year with record temperatures), which does not allow assessing the variability and representativeness of the concentrations and rates measured. This does not take away from the value of the data but weakens conclusions when extrapolating the results, e.g. when stating "Based solely on our data, we estimate a total nitrogen removal of 894 t N d-1 in our study area". This needs to be made clear throughout the paper, including the abstract.

My second concern is the incubation method that is not explained in sufficient detail and that according to the authors did not adequately reproduce the influence of water exchange in the permeable sediments. Ammonification, nitrification and denitrification likely are affected differently through the changes in the water exchange. Since these exchange flows may carry organic matter and oxygen into the sediment, the N-concentrations and conversion rates hinge on this transport and therefore can be altered by changing this flow. The authors pointed this out in the discussion and also added a calculation based on primary production in order to estimate the upper limit of denitrification, nonetheless, the in-situ N-conversion rates therefore may differ significantly from those determined in the lab. This should be made more clear.

The authors may consider including the following references:

Villa et al. (2019) "Benthic nitrogen cycling in the North Sea" Continental Shelf Research.

Kitidis et al. (2017). Seasonal benthic nitrogen cycling in a temperate shelf sea: the Celtic Sea. Biogeochemistry

Detailed comments

Line 16: Dissolved inorganic nitrogen concentration (nitrate, nitrite and ammonium) in the water column showed low levels between 0.2 to 3.2 $\mu$mol L-1. Was the sampling always done at the same daytime?

Line 49: In spite of their putative relevance as an ecosystem service, very little is known about N cycling and N transformation rates in the sediment. Substantial work has been done on sedimentary nitrogen in the North Sea (e.g. see reference lists of the two papers mentioned above) and I suggest softening this statement.

Line 109: cores. . . were incubated in a gas tight batch-incubation setup for 24 hours. The setup of this incubation is critical for the interpretation of the results and the authors need to include more detail: Were all batch incubations done on board or back in the lab? What was the time delay between coring and incubation? What was the length of the sediment cores and were all cores of equal length? What was the height of the water column above the cores, and what were the starting and end oxygen concentrations? Was the water column fully mixed in the core liners? Was stirring the same in all cores? What was the fauna in these cores and how was its activity accounted for? What were the light conditions during incubation? Were all cores incubated with the same water or with the water they came with?

Line 114: $NH_4^+$ and $NO_3^-$ concentration of the added tracer solution was the same as the bottom water concentrations (Tab. 2). This is not clear. Does this mean that you added tracer while maintaining the original nutrient concentration, i.e. removed some water and then added a mix of water and tracer that had the same nutrient concentration as the original water? If that was the case, please specify how much tracer was added and if the different tracer additions (nutrient concentrations differed up to factor 20) could have influenced the incubation experiment. In Line 116 you mention that the label addition was calculated aiming for a maximum enrichment of 5.000 ‰ in substrates and products. How was this achieved if added tracer solution concentration was the same as the bottom water concentrations, and the incubations ran with the same volume of overlaying water?

Table 2: Random frictionless packing in sand produces a porosity of 0.39, and although lower numbers can sometimes be measured, a porosity of 0.29 seems unrealistic. Was the core fully water-saturated? Please check whether these numbers were reported as

weight or volume ratios. Practical salinity is based on ratios and should be expressed by dimensionless number only

Line 118. Upon sampling, incubation water was filtered with a syringe filter (material, manufacturer, 0.45 $\mu$m pore size) Insert material and manufacturer.

Line 157. The surface sediment samples of the cruises HE 383 (06/07.2012) and HE 447 (06.2015) for NOAH-D were analyzed for total carbon and total nitrogen contents with an elemental analyzer (Carlo Erba NA 1500) via gas chromatography calibrated against acetanilide. Please be more specific: How deep was the sediment layer termed "surface sediment"?

Line 184: Three $O_2$ profiles were measured in one sediment core of each station. Please specify the conditions: When exactly where those profiles measured, i.e. how long after retrieval of the core? What were the conditions during the measurements e.g. was there a water layer above the sediment?

Line 195: The lowest oxygen flux was determined at the permeable sediment station NOAH-A with -10.0 mmol m-2 d-1 (Fig. 2), the highest oxygen flux was measured at the impermeable sediment station NOAH-C with -53 mmol m-2 d-1. The semi-permeable sediment station NOAH-D had an oxygen flux of -18.5 to -30.6 mmol m-2 d-1. As pointed out by the authors in the discussion section, the fluxes in the permeable sediment vary with the flow above the sediment. For which flow setting were the fluxes reported here? The same question applies to the statement in Line 203: The lowest ammonification rates were measured in the semi-impermeable sediment at station NOAH-D

Line 284: In total, though, we estimate that benthic N fluxes support between 13 % (at a water depth of 38 m) and 61 % at 10 m depth (Tab. 3) of primary production As this is based on one time summer sampling only, I suggest softening this statement. Line 307: Nitrification rates are relatively independent of permeability, in contrast to ammonification.

This needs further explanation. In the discussion, you mention the potential importance of the flushing of the permeable sediment, which could transport organic matter and oxygen into the sediment. This would have direct implications for both, ammonification as well as nitrification. Why was nitrification relatively independent of permeability?

Line 311: Nitrification rates are lowest at Station NOAH-A. Here, oxygen penetration depth is highest, and the sediment has low organic matter content (Tab. 2), which obviously limits nitrification rates. This statement contradicts the statement on line 307, where you say that "nitrification rates are relatively independent of permeability"

---

## Author Comment (AC1) · 15 Nov 2019

**Reply to the anonymous Referees #1**

(**RC:** Referee Comment; **AR:** Author's Response)

We would like to thank the anonymous Referee #1 for the constructive feedback and thorough review of the manuscript. One main criticism is that the methods, both with regards to incubation and calculation of rates, should be explained in more detail. We agree and will address this issue further in the individual comments. We will also provide an improved description of the method, which should answer many of the reviewers' questions. Besides that, we followed all the suggestions given by the anonymous Referee #1.

**Page 2 line 49-50**

**RC 1:** "very little is known about N cycling and N transformation rates in the sediment". This is a very strong statement and has to be amply justified although I see no evidence of such being the case in the literature. At least, it should be contextually set in a much better way. The justification coming afterwards is rather confusing – are the authors suggesting nothing is known about benthic N cycling in the North Sea because previous studies failed to distinguish between mineralization and nitrification?

**AR:** We agree that this statement was somewhat exaggerated, even though sediment studies of course always deal with the same set of problems, i.e., that the range of sampling designs complicates intercomparisons. In a revised version, we will specify this statement, meaning indeed that ammonification / mineralization is poorly assessed.

**RC 1:** It is suggested that the isotope dilution method employed in this study can 'unravel several N-processes like ammonification, assimilation, nitrification, denitrification (: : :)', DNRA: : :etc within sediments. This requires demonstration, which is not clearly apparent either after this statement or in fact within the rest of the manuscript […].

**AR:** We agree that the isotope dilution method we applied needs to be explained in more detail. Briefly, yes, we did indeed apply the isotope dilution method using parallel enrichments with $^{15}$N-NH$_4^+$ and $^{15}$N-NO$_3^-$ (e.g., Blackburn et al., 1979). This results in a limited number of replicates for individual rates and does indeed affect uncertainty, as the reviewer states. We will clearly state this in the revision. However, we did, precisely for this reason, attempt to assess process rates directly wherever possible, to minimize the problems arising from error propagation.

We will add a new figure as supplementary material to demonstrate which processes were deduced based on concentration changes (net ammonification for ammonium, net nitrification for nitrate concentration), and which rates were measured via tracer addition (gross nitrification and ammonification).
All isotope tracers were indeed added at a tracer level, i.e., site water was replaced by water that contained tracer, but the tracer concentration was adjusted so that it would not change the overall nutrient concentration in the water column.
In theory, it would have been possible to deduce net rates from all incubations, regardless of label additions. The net concentration changes were comparable, but we did not attempt this because gross rates were

determined by difference based on label additions, and we sought to avoid calculating net rates based on 4 cores and gross rates, by difference, based on 2 incubations. As we noted above, we will address the arising uncertainty explicitly in the results and discussion section in a revised manuscript, especially for NOAH-D, where labelled nitrate could, due to difficulties in sediment core retrieval, only be added to one incubation. We hope that by clearly addressing the number of samples and uncertainty, and by clarifying flux calculation (in form of a figure and a more detailed explanation), we have resolved the issues raised by the reviewer.

**General assessment:**

**RC 1:** Propagation of error and significance of differences encountered between different rates at different locations needs to be discussed appropriately, and concrete results presented in this regard. Rates are presented without appropriate attention to this point.

**AR:** As we explained above, the ammonification rates (net and gross) were calculated only with the $^{15}NH_4^+$ tracer sediment cores and the nitrification rates (net and gross) only with the $^{15}NO_3^-$ tracer sediment cores. In a revised version, we will clarify the calculation of the ammonification and nitrification rates including their significance for actual processing, given the uncertainty of measurements.

**RC 1:** Denitrification could be then estimated as the sum of denitrification rates measured in the two cores, and both nitrification and coupled nitrification-denitrification from the $^{15}NH_4^+$ labelled core. Indeed, it seems that was that was done (line 113-115).

**AR**: The denitrification rates are based on MIMS results and are not based on coupled nitrification/denitrification from label incubations. We choose this approach specifically due to the problems that arise from error propagation. If both assessments are compared, however, the rates fall within the same order of magnitude, and are roughly comparable. For the MIMS analysis, the internal precision of the samples was <0.05% for N2/Ar analyses (line153-154). We will clarify this in the revised version.

**RC 1:** $NH_4^+$ concentration in bottom water are usually very low (line 234-235) and how this would affect the accuracy of estimates of nitrification and coupled nitrification-denitrification.

**AR:** We measured only the net and gross ammonification rates using the concentration and isotope ratio of $NH_4^+$ from the $^{15}NH_4^+$ tracer sediment cores. The net and gross nitrification rates were calculated by using the concentration and isotope ratio of $NO_3^-$ from $^{15}NO_3^-$ sediment cores. We will clarify this in the revised version. Accordingly, ammonium concentrations should not have a major effect on the measurement of nitrification rates, because these are based on concentration and isotope label changes in the nitrate pool.
 As we will point out more clearly in a revised version, the addition of label (of any kind) did not change the ambient concentration, because site water was replaced with labelled water that was adjusted to in-situ concentration, so that rates should remain unaltered.

**RC 1:** In Table 3, in the second column, the sum of three process rates is presented with relatively low uncertainty – is this from a flux measurement and therefore this is an aggregate?

**AR:**
In Table 3, we indeed did not address error propagation adequately. We will revise Table 3, considering to exclude assimilation, because this process is poorly assessed in our measurement method, is associated with a relatively large uncertainty, and because rates are small, so that the sum of rates change only a little. We will of course also address uncertainty.

**RC 1:** In NOAH -C (Line 201) to calculate assimilation, the difference between gross and net ammonification alone if propagation of error is accounted for would already be a highly uncertain number: (8.3_2.3 - 6.8_2.3 = 1.5 _ 3.25) and we haven't yet subtracted the gross nitrification rate. The chosen mode of representing uncertainty is also not explained – how is it calculated (looking at the size of the error bars and the spread of results, I presume as the standard error of measurements, with n=2 or 1? Or n=4 or 3?), but has to be made explicit.

**AR:** The assimilation rates were calculated by the sum of gross ammonification minus net ammonification minus gross nitrification rates. It is correct that due to error propagation, rates are highly uncertain In consequence, we decided to skip assimilation from the assessment, because rates are (a) uncertain, and (b) relatively small (based on our measurement). Moreover, we note that our setup as a whole was not ideal to measure assimilation, because cores were incubated in the dark (details on the incubation will be provided in the revised version).

**RC 1:** Reference list seemingly too long compared to actual citations in text.

**AR:** The reference list will be shortened in the next manuscript version. We thoroughly crosschecked the reference list and found one accidental duplicate. All other references in the list are indeed represented by actual citations. However, we do see the point and will, in a revised version, restrict citations to the most relevant ones.

**RC 1:** Present an explicit discussion of the potential pitfalls of using the 15N dilution technique without isotope pairing (Rysgaard, Nielsen et al). In particular, the issues with underestimation of denitrification and nitrification associated with the coupling of both processes and how they could affect interpretation of the results, not mentioned on the manuscript.

**AR:** We will address the method (including its advantages and disadvantages) in more detail in the revision. As in any ex-situ incubation method, the results are not necessarily equal to actual natural rates in the sediment, which we will emphasize in the discussion.
Regarding nitrification and coupled nitrification/denitrification, we would like to point out that our assessment of denitrification is independent of labelling, because it is based on N2 production that was measured by membrane inlet mass spectrometry.

If denitrification removes substantial amounts of nitrate from nitrification, this should decrease the nitrate concentration (i.e., net rate), with no effect on labelling percentage. This may lead to underestimation of nitrification in our assessment, but should also result in a production of $29N_2$ in the MIMS measurement. We will address this in the revision where necessary.

---

## Author Comment (AC2) · 15 Nov 2019

**Reply to the anonymous Referees #2**

(**RC:** Referee Comment; **AR:** Author's Response)

We would like to thank both anonymous Referee #2 for the constructive feedback and thorough review of the manuscript. One main criticism of the reviewer is that the methods, both with regards to incubation and calculation of rates, should be explained in more detail. We agree and will address this issue further in the individual comments. We will also provide an improved description of the method, which should answer many of the reviewers' questions. Besides that, we followed all the suggestions given by the anonymous Referee #2.

**Page 2 line 49**

**RC 2:** In spite of their putative relevance as an ecosystem service, very little is known about N cycling and N transformation rates in the sediment. Substantial work has been done on sedimentary nitrogen in the North Sea (e.g. see reference lists of the two papers mentioned above) and I suggest softening this statement.

**AR:** We agree and will soften this statement in a revised version. We were referring to the fact that ammonification had not taken into consideration, and that we attempt a joint consideration of N-turnover processes, but this statement went a bit too far indeed.

**Page 4 line 109**

**RC 2:** cores. . . were incubated in a gas tight batch-incubation setup for 24 hours. The setup of this incubation is critical for the interpretation of the results and the authors need to include more detail: Were all batch incubations done on board or back in the lab? What was the time delay between coring and incubation? What was the length of the sediment cores and were all cores of equal length? What was the height of the water column above the cores, and what were the starting and end oxygen concentrations? Was the water column fully mixed in the core liners? Was stirring the same in all cores? What was the fauna in these cores and how was its activity accounted for? What were the light conditions during incubation? Were all cores incubated with the same water or with the water they came with?

**AR:** To answer your questions, we will expand the method section and create a new figure of the sediment core incubation for the supplementary material. Briefly, the incubations were done directly after sampling in the ship's laboratory at in-situ temperature. Core incubations were kept in the dark by wrapping cores and overlying water in aluminum foil. All sediment cores were incubated with site water. For faunal activity, we have quantitative assessments based on the observed abundance and biomass of macrofauna from parallel cores.
The water in the incubations was gently stirred, so that the water was mixed, but that surface sediment in the incubations was not resuspended. We will integrate this information in the revised version of the manuscript.

**Page 4 line 114**

**RC 2:** $NH_4^+$ and $NO_3^-$ concentration of the added tracer solution was the same as the bottom water concentrations (Tab. 2). This is not clear. Does this mean that you added tracer while maintaining the original nutrient concentration, i.e. removed some water and then added a mix of water and tracer that had the same nutrient concentration as the original water? If that was the case, please specify how much tracer was added and if the different tracer additions (nutrient concentrations differed up to factor 20) could have influenced the incubation experiment. In Line 116 you mention that the label addition was calculated aiming for a maximum enrichment of 5.000 ‰ in substrates and products. How was this achieved if added tracer solution concentration was the same as the bottom water concentrations, and the incubations ran with the same volume of overlaying water?

**AR:** Yes, we replaced a small volume of site water with label solution, which was (in concentration and volume) adjusted so that no changes in total ammonium or nitrate concentration occurred. As background for the calculation, we used concentration data from previous cruises, assuming that these might be comparable. This assumption was confirmed by subsequent nutrient measurements in the home laboratory, so that we can indeed say that overall nutrient concentration remains unchanged during label addition. We will insert details regarding this in the revised manuscript version.
For the tracer injection, we used two stock solutions of 100 µmol l$^{-1}$ with 50% 15N ($^{15}NH_4^+$ and $^{15}NO_3^-$) in MQ water. At each station, we diluted the stock solution to 2 to 5 µmol l$^{-1}$ ($NH_4^+$ and $NO_3^-$) depending of the station in a 20 ml syringe and injected the appropriate volume into the overlying water (20 cm height above the sediment).

**Page 5 line 118**

**RC 2:** Upon sampling, incubation water was filtered with a syringe filter (material, manufacturer, 0.45 µm pore size) Insert material and manufacturer.

**AR:** We will add the material (cellulose acetate) and manufacturer (Sartorius) of the syringe filter.

**Page 6 line 157**

**RC 2:** The surface sediment samples of the cruises HE 383 (06/07.2012) and HE 447 (06.2015) for NOAH-D were analyzed for total carbon and total nitrogen contents with an elemental analyzer (Carlo Erba NA 1500) via gas chromatography calibrated against acetanilide. Please be more specific: How deep was the sediment layer termed "surface sediment"?

**AR:** "Surface sediment" refers to the first 1 cm. We will add this information in the new manuscript version.

**Page 7 line 184**

**RC 2:** Three $O_2$ profiles were measured in one sediment core of each station. Please specify the conditions: When exactly where those profiles measured, i.e. how long after retrieval of the core? What were the conditions during the measurements e.g. was there a water layer above the sediment?

**AR:** The $O_2$ profiles were measured directly after core retrieval, The time between sampling and O2 profile measurements was always about 10 to 15 minutes. Overlying water column was always adjusted to 20 cm height, and temperature in the temperature-controlled lab was the same as the temperature in the bottom water. We will add this information in the revised manuscript.

**Page 7 line 195**

**RC 2:** The lowest oxygen flux was determined at the permeable sediment station NOAH-A with -10.0 mmol $m^{-2}$ $d^{-1}$ (Fig. 2), the highest oxygen flux was measured at the impermeable sediment station NOAH-C with -53 mmol $m^{-2}$ $d^{-1}$. The semi-permeable sediment station NOAH-D had an oxygen flux of -18.5 to -30.6 mmol $m^{-2}$ $d^{-1}$. As pointed out by the authors in the discussion section, the fluxes in the permeable sediment vary with the flow above the sediment. For which flow setting were the fluxes reported here? The same question applies to the statement in Line 203: The lowest ammonification rates were measured in the semi-impermeable sediment at station NOAH-D.

**AR:** All oxygen fluxes were measured in the sediment core incubations, and gentle stirring was applied, where care was taken not to resuspend the surface sediment.
We are aware of the fact that the rates can, in such an experimental setup, never be identical to the true in situ rates. However, this methodological problem arises for all sediment incubations, and we wanted to demonstrate the relevance and magnitude of ammonification for German Bight sediments – which fits well with rate measurements from previous studies.
However, we will specify the methodological details in a revision.

**Page 10 line 283**

**RC 2:** In total, though, we estimate that benthic N fluxes support between 13 % (at a water depth of 38 m) and 61 % at 10 m depth (Tab. 3) of primary production As this is based on one time summer sampling only, I suggest softening this statement.

**AR:** We will include a statement regarding seasonal variability.

**Page 10 Line 307:**

**RC 2:** Nitrification rates are relatively independent of permeability, in contrast to ammonification. This needs further explanation. In the discussion, you mention the potential importance of the flushing of the permeable sediment, which could transport organic matter and oxygen into the sediment. This would have direct implications for both, ammonification as well as nitrification. Why was nitrification relatively independent of permeability?

**AR:** We found that the sediment permeability and organic matter reactivity affect the ammonification directly, whereas oxygen concentration and ammonification affect the nitrification.

Highest ammonification was measured in impermeable sediments with lowest oxygen penetration depth in sediments, whereas lowest ammonification rates were measured in semi-permeable and permeable sediments with higher oxygen penetration depth. Nitrification is controlled by oxygen availability as well as substrate (ammonium) availability, and we assume that this co-dependence disguises the correlation with permeability (because high oxygen penetration might lead to decreased substrate availability). Overall, permeability of course hast an effect on nitrification, but it is not clearly correlated. We will discuss the complex interplay of sediment characteristics, faunal activity, and resulting N-turnover in a revision.

**Page 11 line 311**

**RC 2:** Nitrification rates are lowest at Station NOAH-A. Here, oxygen penetration depth is highest, and the sediment has low organic matter content (Tab. 2), which obviously limits nitrification rates. This statement contradicts the statement on line 307, where you say that "nitrification rates are relatively independent of permeability".

**AR:** Nitrification rates are affected by oxygen concentration depth in the sediment and by ammonification. At NOAH-A, we measured the highest oxygen concentration depth, which support nitrification, and the lowest organic matter content, which limits the ammonification rate. Here, organic matter turnover indirectly controls nitrification rates (page 11, line 312). Stating that nitrification as such is independent of permeability is indeed not correct – it is poorly correlated in our sample set, but that is of course not the same. We will modify and improve this section in the revised manuscript.

**Page 20 table 2**

**RC 2:** Random frictionless packing in sand produces a porosity of 0.39, and although lower numbers can sometimes be measured, a porosity of 0.29 seems unrealistic. Was the core fully water-saturated? Please check whether these numbers were reported as weight or volume ratios. Practical salinity is based on ratios and should be expressed by dimensionless number only.

**AR:** Regarding porosity at station NOAH-E, the initially stated value of 0.29 is indeed too low. This value was calculated from a sediment sample intended for particle analysis, which was not fully water saturated at the time of measurement. The correct porosity is 0.41 (v/v) and was measured on a fully saturated sample. We will update the manuscript accordingly, and we will change salinity to dimensionless numbers where appropriate.

---

## Author Response (AR1)

**Reply to the anonymous Referees #1 and #2**

(**RC:** Referee Comment; **AR:** Author's Response)

We thank both reviewers for their constructive and detailed feedback on our submitted manuscript. Major issues that were addressed by both reviewers dealt with methodological issues, i.e., clarification of the methods we used, details with regards to error propagation etc.

We realized that the method section was indeed somewhat confused. Moreover, the rate calculation was in some cases based on extrapolations that were not entirely justified. Based on our explanations in the previous response letter, we double-checked the calculations and specifically looked into issues related to error propagation.

In the current manuscript, we took are to uncouple the individual rate measurements as much as possible. Gross rates of nitrification are based on nitrate addition, gross rates of ammonification are based on labelled ammonium addition, both are calculated based on isotope dilution. For the sake of clarity, net rates are now referred to as benthic fluxes, i.e., ammonium, nitrate, or N2 fluxes. These are based on concentration changes in all cores from one station. We rewrote large parts of the method section. As all interpolations in from the previous manuscript were removed, the number of data points is reduced in some cases, and the numbers for benthic fluxes and gross rates changed in some cases. However, the general trends and our overall line of discussion are not affected.

Below, we explain the changes and modifications following the reviewers's suggestions in more detail.

**Referee #1**

**Page 2 line 49-50**

**RC 1:** "very little is known about N cycling and N transformation rates in the sediment". This is a very strong statement and has to be amply justified although I see no evidence of such being the case in the literature. At least, it should be contextually set in a much better way. The justification coming afterwards is rather confusing – are the authors suggesting nothing is known about benthic N cycling in the North Sea because previous studies failed to distinguish between mineralization and nitrification?

**AR:** we softened this statement, now referring specifically to ammonification (line 51 – 59).

**RC 1:** It is suggested that the isotope dilution method employed in this study can 'unravel several N-processes like ammonification, assimilation, nitrification, denitrification (: : :)', DNRA: : :etc within sediments. This requires demonstration, which is not clearly apparent either after this statement or in fact within the rest of the manuscript […].

**AR:** For clarification, we extended the method section. We separated the denitrification measurement more clearly to avoid confusion, prepared a new Figure to demonstrate which fluxes were based on which

measurement, and included a table to the supplementary material that shall enable the reader to follow our flux calculations and gross rate measurements (line 176 – 192).

We also clarified the terminology (replacing net rates by benthic fluxes). We also address the uncertainty of measurements now explicitly in section 4.4 (line 337 – 407).

**General assessment:**

**RC 1:** Propagation of error and significance of differences encountered between different rates at different locations needs to be discussed appropriately, and concrete results presented in this regard. Rates are presented without appropriate attention to this point.

**AR:** As we explained above, the ammonification rates (net and gross) were calculated only with the $^{15}NH_4^+$ tracer sediment cores and the nitrification rates (net and gross) only with the $^{15}NO_3^-$ tracer sediment cores. in the revised version, we put this more clearly (changing the method section and adding a new figure – section 4.4 – line 337 - 407). All error bars are standard deviations of measurements. Individual rate measurements and fluxes are uncoupled to minimize problems arising from error propagation, this should be well-represented in Fig. 5.

**RC 1:** Denitrification could be then estimated as the sum of denitrification rates measured in the two cores, and both nitrification and coupled nitrification-denitrification from the $^{15}NH_4^+$ labelled core. Indeed, it seems that was that was done (line 113-115).

**AR**: The denitrification rates are based on MIMS results and are not based on coupled nitrification/denitrification from label incubations. We choose this approach specifically due to the problems that arise from error propagation. In the discussion section, we now also address that denitrification (measured by MIMS), measured nitrate fluxes and nitrification match in our sampled sediment cores, so that the calculated budget is closed (line 132 – 137)

**RC 1:** $NH_4^+$ concentration in bottom water are usually very low (line 234-235) and how this would affect the accuracy of estimates of nitrification and coupled nitrification-denitrification.

**AR:** We measured only the net and gross ammonification rates using the concentration and isotope ratio of $NH_4^+$ from the $^{15}NH_4^+$ tracer sediment cores. We clarified this in the revised version (see especially new Figure 2). As we stated in the initial response letter, ammonium concentrations should then not have a major effect on the measurement of nitrification rates, because these are based on concentration and isotope label changes in the nitrate pool.

In the method section, we now also point out that the addition of label (of any kind) did not change the ambient concentration, because site water was replaced with labelled water that was adjusted to in-situ concentration, so that rates should remain unaltered (lin1 108 – 130).

**RC 1:** In Table 3, in the second column, the sum of three process rates is presented with relatively low uncertainty – is this from a flux measurement and therefore this is an aggregate?

**AR:**
We excluded assimilation (due to high uncertainty) from Table 3 and from the manuscript.

**RC 1:** In NOAH -C (Line 201) to calculate assimilation, the difference between gross and net ammonification alone if propagation of error is accounted for would already be a highly uncertain number: (8.3_2.3 - 6.8_2.3 = 1.5 _ 3.25) and we haven't yet subtracted the gross nitrification rate. The chosen mode of representing uncertainty is also not explained – how is it calculated (looking at the size of the error bars and the spread of results, I presume as the standard error of measurements, with n=2 or 1? Or n=4 or 3?), but has to be made explicit.

**AR:** It is correct that due to error propagation, rates are highly uncertain In consequence, we decided to skip assimilation from the assessment. Moreover, we note that our setup as a whole was not ideal to measure assimilation, because cores were incubated in the dark (now added in the method section – line 176 – 192).

**RC 1:** Reference list seemingly too long compared to actual citations in text.

**AR:** The reference list will be shortened in the next manuscript version. We thoroughly crosschecked the reference list and found one accidental duplicate. All other references in the list are indeed represented by actual citations. However, we do see the point and will, in a revised version, restrict citations to the most relevant ones.

**RC 1:** Present an explicit discussion of the potential pitfalls of using the $^{15}$N dilution technique without isotope pairing (Rysgaard, Nielsen et al). In particular, the issues with underestimation of denitrification and nitrification associated with the coupling of both processes and how they could affect interpretation of the results, not mentioned on the manuscript.

**AR:** We extended the method section to clarify our calculation procedures for gross rates of nitrification, ammonification, benthic fluxes of nitrate and ammonium, and $N_2$ production. We hope that this resolves the methodological issues mentioned here (section 2.3).

**Referee #2**

**Page 2 line 49**

**RC 2:** In spite of their putative relevance as an ecosystem service, very little is known about N cycling and N transformation rates in the sediment. Substantial work has been done on sedimentary nitrogen in the North Sea (e.g. see reference lists of the two papers mentioned above) and I suggest softening this statement.

**AR:** In the revised version, we softened the statement and now explicitly refer to ammonification (line 51 – 59).

**Page 4 line 109**

**RC 2:** cores. . . were incubated in a gas tight batch-incubation setup for 24 hours. The setup of this incubation is critical for the interpretation of the results and the authors need to include more detail: Were all batch incubations done on board or back in the lab? What was the time delay between coring and incubation? What was the length of the sediment cores and were all cores of equal length? What was the height of the water column above the cores, and what were the starting and end oxygen concentrations? Was the water column fully mixed in the core liners? Was stirring the same in all cores? What was the fauna in these cores and how was its activity accounted for? What were the light conditions during incubation? Were all cores incubated with the same water or with the water they came with?

**AR:** We expanded the method section and created a new figure of the core incubation setup (Fig. 2). We also added information regarding stirring and core handling etc. to the manuscript (line 108 – 130).

**Page 4 line 114**

**RC 2:** $NH_4^+$ and $NO_3^-$ concentration of the added tracer solution was the same as the bottom water concentrations (Tab. 2). This is not clear. Does this mean that you added tracer while maintaining the original nutrient concentration, i.e. removed some water and then added a mix of water and tracer that had the same nutrient concentration as the original water? If that was the case, please specify how much tracer was added and if the different tracer additions (nutrient concentrations differed up to factor 20) could have influenced the incubation experiment. In Line 116 you mention that the label addition was calculated aiming for a maximum enrichment of 5.000 ‰ in substrates and products. How was this achieved if added tracer solution concentration was the same as the bottom water concentrations, and the incubations ran with the same volume of overlaying water?

**AR:** As we outlined in the original response letter, we replaced site water with label solution. More detail regarding this addition has been added to the method section (line 109 – 130).

**Page 5 line 118**

**RC 2:** Upon sampling, incubation water was filtered with a syringe filter (material, manufacturer, 0.45 µm pore size) Insert material and manufacturer.

**AR:** Done (line 126 – 128).

**Page 6 line 157**

**RC 2:** The surface sediment samples of the cruises HE 383 (06/07.2012) and HE 447 (06.2015) for NOAH-D were analyzed for total carbon and total nitrogen contents with an elemental analyzer (Carlo Erba NA

1500) via gas chromatography calibrated against acetanilide. Please be more specific: How deep was the sediment layer termed "surface sediment"?

**AR:** Top 1 cm – added to the manuscript (line 145).

**Page 7 line 184**

**RC 2:** Three $O_2$ profiles were measured in one sediment core of each station. Please specify the conditions: When exactly where those profiles measured, i.e. how long after retrieval of the core? What were the conditions during the measurements e.g. was there a water layer above the sediment?

**AR:** We added more detail to the method section (line 139 - 143).

**Page 7 line 195**

**RC 2:** The lowest oxygen flux was determined at the permeable sediment station NOAH-A with -10.0 mmol $m^{-2}$ $d^{-1}$, the highest oxygen flux was measured at the impermeable sediment station NOAH-C with -53 mmol $m^{-2}$ $d^{-1}$. The semi-permeable sediment station NOAH-D had an oxygen flux of -18.5 to -30.6 mmol $m^{-2}$ $d^{-1}$. As pointed out by the authors in the discussion section, the fluxes in the permeable sediment vary with the flow above the sediment. For which flow setting were the fluxes reported here? The same question applies to the statement in Line 203: The lowest ammonification rates were measured in the semi-impermeable sediment at station NOAH-D.

**AR:** We amended the method section (line 176 – 192).

**Page 10 line 283**

**RC 2:** In total, though, we estimate that benthic N fluxes support between 13 % (at a water depth of 38 m) and 61 % at 10 m depth (Tab. 3) of primary production As this is based on one time summer sampling only, I suggest softening this statement.

**AR:** We now address seasonal effects in the discusson section 4.2 (denitrification). We also rewrote the discussion, now addressing the significance of benthic fluxes in a separate section, 4.4 (line 337 – 407)

**Page 10 Line 307:**

**RC 2:** Nitrification rates are relatively independent of permeability, in contrast to ammonification. This needs further explanation. In the discussion, you mention the potential importance of the flushing of the permeable sediment, which could transport organic matter and oxygen into the sediment. This would have

direct implications for both, ammonification as well as nitrification. Why was nitrification relatively independent of permeability?

**AR:** Nitrification is controlled by the availability of oxygen and nitrate, and thesubstrate limitation apparently limits nitrification at the stations with higher sediment permeability. We discuss the regulation of nitrification now in the revised discussion (line 271 – 306).

**Page 11 line 311**

**RC 2:** Nitrification rates are lowest at Station NOAH-A. Here, oxygen penetration depth is highest, and the sediment has low organic matter content (Tab. 2), which obviously limits nitrification rates. This statement contradicts the statement on line 307, where you say that "nitrification rates are relatively independent of permeability".

**AR:** As outlined above, we now discuss the regulation of nitrification in our sample set in more detail (line 271 – 306).

**Page 20 table 2**

**RC 2:** Random frictionless packing in sand produces a porosity of 0.39, and although lower numbers can sometimes be measured, a porosity of 0.29 seems unrealistic. Was the core fully water-saturated? Please check whether these numbers were reported as weight or volume ratios. Practical salinity is based on ratios and should be expressed by dimensionless number only.

**AR** As we outlined in the initial letter, this low value was an artifact, we now corrected it (new: 0.41). We also now refer to salinity as a dimensionless number. The correct porosity is 0.41 (v/v) and was measured on a fully saturated sample. We updated the manuscript accordingly, and we changed the salinity to dimensionless numbers where appropriate (Tab. 2).

---

## Author Response (AR3)

**Response Letter to Associate Editor**

Dear Prof. Perran Cook,

Thank you for your additional comments to further improve our manuscript. Please find below our specific responses to your comments.

| Editor's comment | Our response |
|---|---|
| Line 124 – 5.000 or 5‰ ?  NEW: Line 123 | We labeled with $^{15}NH_4^+$ and $^{15}NO_3^-$ to get a final enrichment of 5,000‰ in the sediment core. We changed the point to a comma: "The label addition was calculated aiming for a maximum enrichment of 5,000 ‰ in substrates." |
| Line 214 – Please elaborate why the requirements for the IPT were not met.  New: Line 213 - 214 | The isotope pairing technique is a widely used method to asses contributions of $NO_3^-$ reducing processes, where labeled $^{15}NO_3^-$ (99.9 atom%) is initiated to the overlying water to permit the production of $^{15}N-N_2$. We labeled with much lower $^{15}NO_3^-$ (5,000 ‰ in the overlying water of the sediment core) to calculate the gross nitrification rates and to be inside the measuring range of the used IRMS. The enrichment of $^{15}NO_3^-$ is too low to measure any $^{15}N-N_2$ species. We rewrote the section: "…were not met because the labeled $^{15}NO_3^-$ in the overlying water is too low to measure any $^{15}N-N_2$ species." |
| Line 204 as for the ammonification, (not likewise to…)  NEW: Line 202 | We rewrote the sentence as you mentioned. |

Kind regards,

Alexander Bratek, Justus van Beusekom, Andreas Neumann, Tina Sanders, Jana Friedrich, Kay-Christian Emeis and Kirstin Dähnke